# InteractBench: Benchmarking LLMs on Competitive Programming under Unrevealed Information

**Jiaze Li** [1 2 3]  **Aocheng Shen** [1]  **Bing Liu** [1]  **Boyu Zhang** [1]  **Xiaoxuan Fan** [1]  **Qiankun Zhang** [1 2 3]  **Xianjun Deng** [1 3]

## Abstract

Competitive programming is increasingly being used to evaluate the algorithmic reasoning capabilities of large language models (LLMs). However, existing benchmarks primarily focus on full-information tasks where all problem inputs are provided upfront. This overlooks a critical dimension of algorithmic reasoning: the ability of generated programs to operate when key information is not revealed upfront. *Interactive* problems, a distinctive component of competitive programming, embody this challenge. These problems require programs to engage in multi-round interaction with an interactor (a judge program) under strict protocol constraints and limited query budgets, with new information revealed *only* in response to queries. To address this gap, we introduce *InteractBench*, a benchmark comprising 322 high-quality interactive problems curated from Codeforces, AtCoder, IOI, and ICPC. Each problem is packaged with executable local interactors, enabling fully offline evaluation. Unlike existing benchmarks, InteractBench assesses whether model-generated code can acquire information and track state dynamically. Our evaluation reveals a significant interaction gap: even the most advanced reasoning models achieve limited success on interactive problems. Beyond success rates, we propose a fine-grained failure taxonomy to diagnose the root causes of these deficiencies. Although algorithmic logic errors remain dominant, protocol violations and query-budget overruns are frequent. Code is available at https://github.com/kmsgk0/InteractBench.

## 1. Introduction

Frontier large language models (LLMs) have made rapid progress in code generation and algorithmic reasoning. On widely used benchmarks such as HumanEval (Chen et al., 2021) and MBPP (Austin et al., 2021), performance has approached saturation, making it difficult to meaningfully differentiate top models using a single overall success rate.

Competitive programming pairs formally specified algorithmic tasks with automated judging, providing reliable correctness verdicts under strict time and memory limits, and has been argued to serve as an effective evaluator for LLMs (Huang et al., 2024). This has motivated multiple competitive-programming benchmarks, including CodeContests (Li et al., 2022), LiveCodeBench (Jain et al., 2024), and USACO (Shi et al., 2024). More recently, benchmarks such as CodeElo (Quan et al., 2025), HLCE (Li et al., 2025), ICPC-Eval (Xu et al., 2025), LiveCodeBench Pro (Zheng et al., 2025), and LiveOIBench (Zou et al., 2025) have emerged. These benchmarks suggest progress on harder competitive-programming problems, particularly with reasoning-focused models. Yet most adopt a full-information evaluation protocol: the solver reads a complete, static input at the start of execution; such tasks are referred to as *batch-style* problems. As a result, it remains unclear how well model-generated programs can solve *interactive* problems when key information is not revealed upfront and must be acquired through runtime queries.

**Batch-style and interactive problems.** The *batch-style* problem has all the input provided upfront, and the input does not depend on the solver's behavior (Verhoeff, 2009). In an *interactive* problem, key information is held by an interactor, the program responsible for answering queries (Mirzayanov, 2016; CMS Development Team, 2017). This information is revealed only in response to the solver's queries, subject to a strict protocol and a problem-specific query budget.[1] Success therefore depends on an online querying strategy in addition to producing a correct final answer. Figure 1 illustrates this distinction with an example problem.

---

[1]School of Cyber Science and Engineering, Huazhong University of Science and Technology, Wuhan, China [2]Key Laboratory of Cyberspace Security, Ministry of Education, Zhengzhou, China [3]Hubei Key Laboratory of Distributed System Security, Wuhan, China. Correspondence to: Qiankun Zhang <qiankun@hust.edu.cn>.

*Proceedings of the 43rd International Conference on Machine Learning*, Seoul, South Korea. PMLR 306, 2026. Copyright 2026 by the author(s).

[1]This differs from HLCE (Li et al., 2025)'s use of "interactive-based" to denote function-signature evaluation.

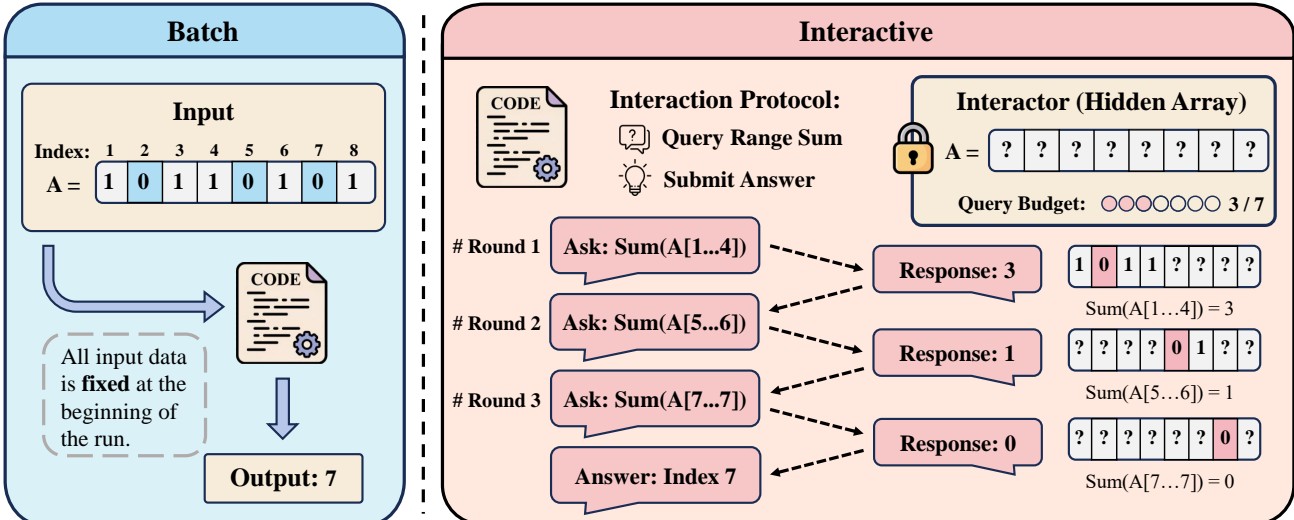

*Figure 1.* An illustration of the difference between batch-style and interactive problems. We use "Find the K-th Zero" as an example. In the batch-style setting (left), the program receives the complete input array at the start of execution and outputs the answer in a single shot. In the interactive setting (right), the array is hidden and the program must interact with an interactor (judge program) under a strict protocol and a limited query budget to gather information. The program can make several queries that ask the sum of all elements in a specific interval, and receive the answer from the interactor. Each query consumes part of the budget, and the interactor's response depends on the query and the hidden input. The program reasons over multiple rounds before submitting the final answer.

These interactive settings introduce additional evaluation challenges; in particular, the following limitations and challenges in existing benchmarks need to be addressed:

- **Interactive coverage.** Many benchmarks focus on batch-style evaluation (Jain et al., 2024; Shi et al., 2024; Xu et al., 2025), and interactive problems are rare or explicitly excluded to simplify evaluation infrastructure.
- **Offline evaluation barriers.** Some evaluations, such as CodeElo (Quan et al., 2025) and HLCE (Li et al., 2025), depend on remote submissions to external online judges, which hinders fully offline evaluation and large-scale experimentation.
- **Limited outcome analysis.** Many studies primarily report an overall success metric (e.g., pass@k) and provide limited diagnostics for distinguishing algorithmic errors from interaction-specific failures (Quan et al., 2025; Li et al., 2025; Zheng et al., 2025; Zou et al., 2025).

To address these limitations, we introduce *InteractBench*, a benchmark of competitive-programming interactive problems, together with an evaluation harness to run these interactions reliably offline. Our contributions are as follows:

**Contribution #1: a benchmark of interactive problems.** We curate interactive problems from Codeforces (Codeforces, 2026), AtCoder (AtCoder Inc., 2026), IOI (Interna-

tional Olympiad in Informatics, 2026), and ICPC (ICPC Foundation, 2026). Each problem is annotated with consolidated categories and difficulty tiers. We release Interact-Bench as versioned snapshots with semi-annual refreshes to maintain freshness, and report a time-split temporal contamination diagnostic in Appendix C.3.

**Contribution #2: a self-contained evaluation harness.** Each problem is packaged with an executable local interactor, enabling fully offline evaluation without external judge submission and supporting repeatable experimentation.

**Contribution #3: interaction-aware diagnostics.** Beyond pass@k, our harness checks protocol compliance, enforces query budgets, and reports a failure taxonomy that distinguishes algorithmic errors such as wrong answers from interaction-specific failures such as protocol violations, query-budget overruns, and deadlocks or timeouts.

## 2. Related Work

We discuss related work along two axes: general code benchmarks and competitive programming benchmarks. Table 1 compares representative benchmarks.

**General code benchmarks.** Early execution-based benchmarks such as HumanEval (Chen et al., 2021) and MBPP (Austin et al., 2021) focus on short, self-contained function synthesis, where performance has become near-saturated for state-of-the-art models. Recent benchmarks broaden to harder tasks (Hendrycks et al., 2021; Jain et al., 2024) and software engineering (Jimenez et al., 2023). Other lines of work strengthen test suites (Liu et al., 2023a), cover complex instructions and diverse function calls (Zhuo et al., 2024), use explicit test-case generation (He et al., 2025; Wang et al., 2025b), and expand to domain-specific and repository-level benchmarks such as DS-1000 (Lai et al., 2023), RepoBench (Liu et al., 2023b), and Cross-CodeEval (Ding et al., 2023). Despite these advances, most benchmarks still score one-shot submissions on fixed tests. Recent agent-style programming benchmarks, such as SWE-bench (Jimenez et al., 2023), AgentBench (Liu et al., 2023c), InterCode (Yang et al., 2023), and CodeAct (Wang et al., 2024), emphasize the *development loop*, where a model iteratively edits code with execution feedback (e.g., via tools and tests). However, few of these benchmarks cover interactive problems that require a strict multi-round protocol and an explicit query budget. InteractBench targets a different setting: the LLM generates a self-contained program once, and success depends on whether the generated program implements a sound interactive strategy during program–interactor interaction, rather than on iterative debugging.

**Competitive programming benchmarks.** Competitive programming tasks provide a demanding test for algorithmic reasoning under strict time and memory constraints. Representative benchmarks include AlphaCode (Li et al., 2022), ICPC-Eval (Xu et al., 2025), LiveCodeBench (Jain et al., 2024), LiveCodeBench Pro (Zheng et al., 2025), and LiveOIBench (Zou et al., 2025). Many focus on batch-style problems where the full input is provided upfront (Shi et al., 2024; Xu et al., 2025; Jain et al., 2024). Some rely on submissions to external online judges (Quan et al., 2025; Li et al., 2025), limiting fully offline evaluation. Even when interactive problems are included, they typically constitute a small subset (Zheng et al., 2025; Zou et al., 2025), and most suites provide limited protocol-level diagnostics beyond pass/fail outcomes. Other suites and analyses include OJBench (Wang et al., 2025c) and AetherCode (Wang et al., 2025a). InteractBench addresses these limitations by providing offline-executable local interactors and protocol-aware diagnostics for interactive problems.

## 3. InteractBench Construction

**Overview.** Unlike in batch-style evaluation, interactive problems require multi-round interaction under strict protocols and explicit query budgets. To enable self-contained offline evaluation, we package each task with an *executable*

*local interactor* and a sandboxed harness. The harness records interaction transcripts and structured termination reasons, enabling interaction-aware diagnosis. We begin with problem definition and task selection, then describe an execution-driven propose–validate–adjudicate loop for constructing offline evaluator artifacts, and finally the post-hoc online agreement audit (Figure 2). Appendix D.3 provides an end-to-end packaged task example.

### 3.1. Problem Definition

Following standard interactive judging conventions (Mirzayanov, 2016; CMS Development Team, 2017), we formalize the competitive-programming interactive tasks considered in InteractBench as follows.

Let $S$ be the solver, $J$ be the interactor, $x$ be the hidden test case, $\Pi$ be the interaction protocol, and $B$ be the query budget. For an execution that terminates after $T$ rounds under $\Pi$, let $q_t$ and $r_t$ denote the solver's query and the interactor's response at round $t$, respectively. We use $\tau_t$ to denote the transcript after the first $t$ rounds, starting from an initial public transcript $\tau_0$ (empty if no public data is revealed), and write $\tau_{<t} := \tau_{t-1}$ for the transcript prefix before round $t$.

An execution then unfolds as

$$
\begin{aligned}
q_t &= S_{\mathrm{qry}}(\tau_{<t}), \\
r_t &= J(x, \tau_{<t}, q_t), \\
\tau_t &= \tau_{<t} \circ (q_t, r_t), \qquad t = 1, \ldots, T, \\
\tau &= \tau_T, \\
a &= S_{\mathrm{out}}(\tau), \\
v &= \mathrm{Eval}(a, \tau, x; \Pi, B) \in \{\mathrm{Accepted}, \mathrm{Rejected}\}.
\end{aligned}
$$

Here $\circ$ denotes transcript concatenation, $S_{\mathrm{qry}}$ is the query-generation component of $S$, and $S_{\mathrm{out}}$ is its final-output component. Thus $\tau$ is the complete interaction transcript. The protocol $\Pi$ specifies message formats and termination conditions, and induces a budgeted count $Q_\Pi(\tau)$ (e.g., queries, rounds, or grader calls). The evaluator $\mathrm{Eval}$ returns Accepted if and only if $\tau$ conforms to $\Pi$, $Q_\Pi(\tau) \leq B$, and the final output $a$ is correct for $x$; otherwise it returns Rejected.

### 3.2. Task Selection

We curate candidates from Codeforces, AtCoder, IOI, and ICPC. A task is included only if it satisfies all of the following criteria:

- **Multi-round interaction.** The solution must interact with an external interactor for multiple rounds rather than producing a one-shot output.
- **Protocol constraints.** The interaction protocol is sufficiently well-defined (e.g., it specifies message formats

*Table 1.* Comparison of representative code and competitive programming benchmarks. Columns report the number of problems; whether the benchmark has interactive support (with counts when available); interface type (*stdio* or grader-linked (CMS Development Team, 2017)); and whether it provides self-contained evaluation, as well as tags and difficulty. Interactive counts may vary for continuously updated benchmarks.

| Benchmark | # Problems | Interactive | Interface | Self-contained | Tags | Difficulty |
|---|---|---|---|---|---|---|
| HumanEval (Chen et al., 2021) | 164 | ✗ | – | ✓ | ✗ | ✗ |
| MBPP (Austin et al., 2021) | 974 | ✗ | – | ✓ | ✗ | ✗ |
| APPS (Hendrycks et al., 2021) | 10,000 | ✗ | – | ✓ | ✗ | ✓ |
| CodeContests (Li et al., 2022) | 13,328 | ✗ | – | ✓ | ✓ | ✓ |
| LiveCodeBench (Jain et al., 2024) | 511 | ✗ | – | ✓ | ✗ | ✓ |
| USACO (Shi et al., 2024) | 307 | ✗ | – | ✓ | ✗ | ✓ |
| ICPC-Eval (Xu et al., 2025) | 118 | ✗ | – | ✓ | ✓ | ✗ |
| HLCE (Li et al., 2025) | 235 | ✓ | grader | ✗ | ✗ | ✗ |
| LiveCodeBench Pro (Zheng et al., 2025) | 584 | ✓ (21) | stdio | ✓ | ✓ | ✓ |
| LiveOIBench (Zou et al., 2025) | 403 | ✓ (23) | stdio+grader | ✓ | ✓ | ✓ |
| CodeElo (Quan et al., 2025) | 408 | ✓ (17) | stdio | ✗ | ✓ | ✓ |
| **InteractBench (Ours)** | **322** | ✓ (322) | stdio+grader | ✓ | ✓ | ✓ |

and termination conditions) to allow offline execution and compliance checking.

- **Query budget.** The task has an explicit or enforceable budget, such as the number of queries, rounds, or calls, that can be monitored and validated during execution.

InteractBench covers both stdio and grader-linked interfaces. For the latter, we retain only multi-query settings with an explicit budget and exclude essentially batch-style single-call interfaces.

**Difficulty and categories.** Each task is annotated with a difficulty tier (Easy/Medium/Hard) and one or more category labels. Experts assign difficulty based on the problem statement and expected solution complexity, consulting contest statistics when available. Category labeling starts from platform metadata, is consolidated into seven categories, and is verified by experts. When platform labels are missing or inconsistent, model-assisted suggestions provide candidate labels before expert finalization. The seven categories are Graph, Search, Greedy, Bit, Data Structures, Math, and Game. Graph refers to graph or tree structure, Search to feasible-set localization, Greedy to committed step-wise construction, and Bit to Boolean/parity-style feedback or compact encodings. Data Structures marks tasks driven by structured state, such as orders, partitions, or reconstruction invariants. Math covers arithmetic, algebraic, or geometric deduction, while Game denotes minimax or worst-case strategic reasoning. Appendix B.1 provides more detailed category definitions.

### 3.3. Construction Pipeline

We construct offline evaluator artifacts via an execution-driven propose–validate–adjudicate loop. *Proposer* models draft candidate generators and interactors from the problem statement. Candidates are then *validated* against a per-task verification pool of submissions with known official verdicts. Failed candidates receive trace-grounded feedback from an *adjudicator* model and re-enter the loop; remaining cases are escalated to human experts after a fixed iteration cap.

**Propose.** Constructing reliable interactive evaluators requires jointly designing a test case generator and a judge-side interactor. We adopt an execution-driven propose–validate–adjudicate loop with two proposer models, GPT-5.2 (OpenAI, 2025a) and Gemini-3-Pro-Preview (Deep-Mind, 2025); we use GPT-5.2 as a code-oriented adjudicator model, while acceptance is determined by execution-based validation. First, the proposer models draft candidate generators, and the adjudicator model selects one that satisfies the contract. We then generate a fixed, seeded suite of 100 test cases per task. Based on the selected generator and input schema, the proposer models draft candidate interactors.

**Validate.** We validate candidate evaluator artifacts for fidelity to official verdicts and strict enforcement of the protocol and budget constraints. Our offline harness additionally records a structured rejection reason from a unified failure taxonomy (Section 4.2) for diagnosis and analysis. Interactor mode is determined by the original problem statement. We implement an adaptive interactor only when the statement permits history-dependent choices by the interactor. In such tasks, the interactor may select a hidden value consistent with the transcript and choose among multiple statement-permitted replies to make identification harder; see Appendix D.4 for a concrete example. Otherwise, we use a non-adaptive interactor where the hidden test case is fixed before interaction begins.

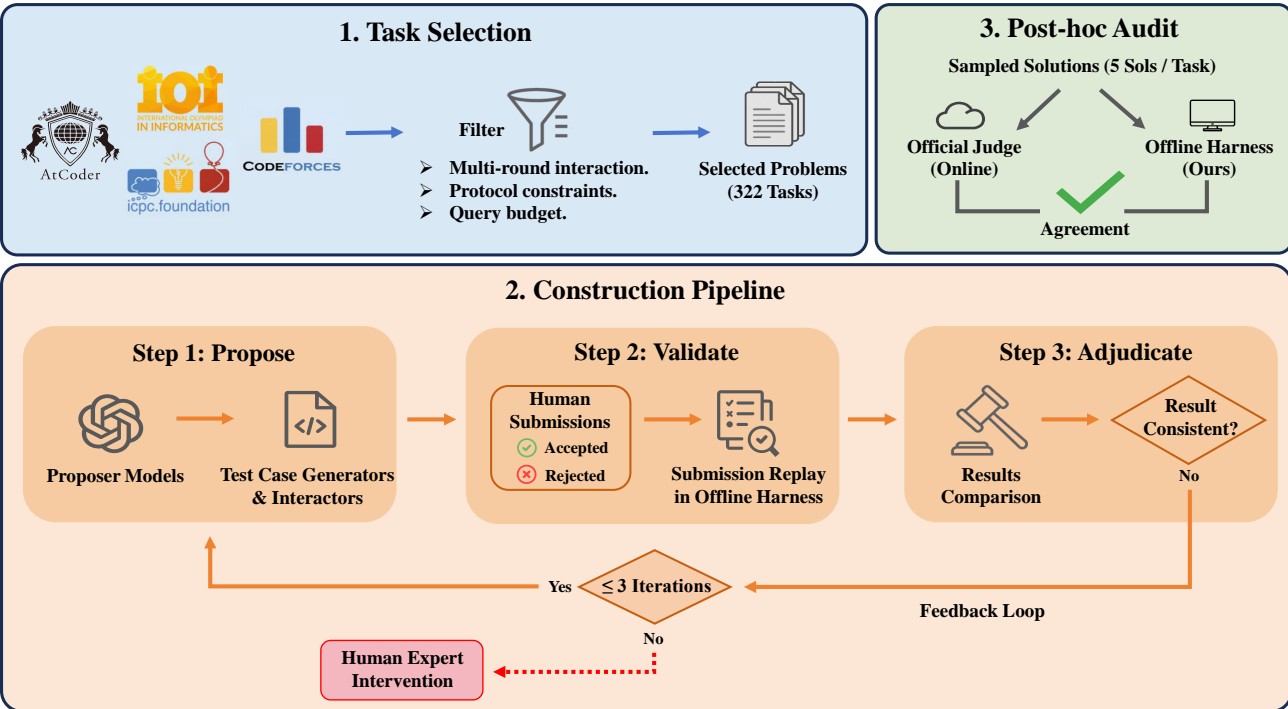

*Figure 2.* An overview of the InteractBench construction pipeline. We select interactive tasks with multi-round protocols and explicit query budgets and construct offline evaluator artifacts (generator and local interactor) via a propose–validate–adjudicate loop validated on a per-task verification pool. Finally, where permitted, we post-hoc audit agreement by submitting a small sample of solutions to the official evaluation interface and comparing Accepted/Rejected outcomes with our offline harness.

We build a per-task verification pool with 15 accepted submissions and 30 rejected submissions. The rejected side includes 15 rejected human submissions with known official judge outcomes and 15 mutation-derived rejected probes obtained by lightly mutating accepted or rejected submissions. These mutations target common failure surfaces (e.g., protocol/format deviations, query-budget violations, and minor logic perturbations) that a faithful evaluator should reject. A candidate is accepted only if our offline harness matches the official judge outcomes on all human submissions and rejects all mutation-derived probes. Appendix D.5 details the probe construction. When official evaluation packages are publicly available (e.g., IOI test suites and checkers/graders/managers), we adapt them to our harness.

Validation is performed fully offline: the harness runs the solver and a local interactor as two isolated processes connected via standard streams. It enforces protocol and budget constraints via explicit counters and logs the interaction transcript and termination signals for diagnosis. Both the solver and the interactor are sandboxed with explicit per-process time and memory budgets; by default, we follow the original task resource budgets.

This two-process interface makes evaluation portable across machines and, for stdio tasks, across programming languages under the same packaged interactor, hidden test cases, and sandbox configuration. Portability supports repeatable experimentation without dependence on external judging infrastructure. In particular, cross-language execution requires only that the solver follows the protocol, including explicit flushing after each query (Mirzayanov, 2016).

**Adjudicate.** Candidates that fail validation proceed to adjudication: we provide trace-grounded feedback and run another iteration. We cap the loop at three iterations before escalating remaining ambiguous cases to human experts for final auditing and correction. After reaching the iteration cap, 31 tasks required expert intervention. Appendix D.6 provides a representative expert-intervention case study. Our human experts are experienced competitive programmers with strong backgrounds in algorithms and interactive judging, and they finalize the evaluator by revising the generator and interactor based on trace evidence.

### 3.4. Post-hoc Audit

As a supplementary sanity check, we conduct a one-time post-hoc online agreement audit by manually submitting sampled model-generated solutions to the official evaluation interface, where permitted. Direct submission-based evaluation does not scale well: platforms such as Codeforces

prohibit automated submissions and impose strict rate limits (Zou et al., 2025; Xu et al., 2025). We therefore restrict this audit to 5 submissions per task and verify consistent Accepted/Rejected outcomes between the official evaluation interface and our offline harness. Submissions are made conservatively in non-competitive settings.

## 4. Experiments

### 4.1. Experimental Setup

**Overview.** We study interactive problem solving under *unrevealed information at execution time*. Solvers must acquire information through multi-round queries under strict protocol constraints and query budgets. We conduct three analyses. In Experiment #1, we report success rates stratified by difficulty (Table 2). In Experiment #2, we break down performance across categories that require *information acquisition* and *dynamic state tracking* under partial observability (Table 3). In Experiment #3, we diagnose whether failures stem from algorithmic logic errors (WA) or interaction-specific breakdowns such as protocol violations and query-budget overruns.

**Tasks.** We report results on InteractBench. Each task is evaluated locally under our harness with its executable local interactor (Section 3). For analysis in Experiments #1–2, tasks are annotated with difficulty tiers and categories; dataset statistics are provided in Appendix B.1.

**Models and outputs.** We evaluate a mix of closed-source and open-source LLMs that generate self-contained programs. Our suite includes reasoning configurations (e.g., DeepSeek-V3.2-Thinking (Liu et al., 2025), GPT-5.2 (OpenAI, 2025a), Gemini-3-Pro-Preview (DeepMind, 2025), Gemini-2.5-Pro (Google DeepMind, 2025b), Gemini-2.5-Flash (Google DeepMind, 2025a), Claude Opus 4.5 (Anthropic, 2025), and GPT-OSS (Agarwal et al., 2025)) as well as standard non-reasoning baselines (DeepSeek-V3.2, GPT-4.1 (OpenAI, 2025b), Qwen3-32B-NonThinking, and Qwen3-14B-NonThinking). In total, we evaluate 16 model configurations (12 reasoning and 4 non-reasoning). Detailed model information is provided in Appendix B.2. Since some model families have both a reasoning and a non-reasoning variant (notably DeepSeek-V3.2, Qwen3-32B, and Qwen3-14B), we report each variant as a separate configuration and use within-family comparisons to isolate the effect of explicit reasoning. While the harness is language-agnostic, we report results for C++ submissions in this paper to match common competitive-programming practice and prior benchmarks (Zou et al., 2025; Xu et al., 2025) (Appendix C.4 reports cross-language and template comparisons).

**Prompting and sampling.** We use a standardized zero-shot prompt consisting of the problem statement plus explicit instructions for interactive I/O (e.g., adhere to the protocol and flush after each query for stdio tasks). For model access, we query closed-source models via provider APIs. For open-source checkpoints, we self-host those that fit on our inference server (8 NVIDIA A100 GPUs); otherwise, we access them via provider APIs. For each problem, we sample $n = 10$ independent solutions with temperature 0.8 to reduce the variance of pass@k estimates. Appendix B.3 lists the evaluation-time prompt wrapper.

**Execution environment.** We compile and execute all submissions in an isolated sandbox (Docker with cgroups) on Ubuntu 22.04 with an Intel Xeon Platinum 8470Q CPU. We enforce per-run limits of 2 CPU cores and 4 GB RAM. Solutions are compiled with g++ 11.4.0 (C++17, -O2); we use Python 3.12, Java 21, and Go 1.21 for cross-language evaluation (Appendix C.4). We enforce time limits using CPU time (to align with competitive-programming judging), together with a conservative wall-clock cap to terminate hanging runs such as blocked I/O or deadlocks.

### 4.2. Metrics

**pass@k.** We report pass@1 and pass@5. Given $n$ sampled solutions for a problem, let $c$ denote the number of solutions accepted on the full hidden test suite under our harness. Following HumanEval (Chen et al., 2021), we estimate pass@k using the unbiased estimator:

$$\text{pass@}k = 1 - \frac{\binom{n-c}{k}}{\binom{n}{k}}.$$

We then average pass@k over the evaluation set.

**Failure taxonomy.** To support interaction-aware diagnosis, we assign each unsuccessful execution a *primary* failure label (mutually exclusive): **IDLE** (blocked I/O or deadlock terminated by the wall-clock cap), **PE** (protocol/format violations), **QLE** (query-budget overrun), **CE** (compile error), **RE** (runtime error), **TLE** (time limit exceeded), **MLE** (memory limit exceeded), and **WA** (protocol-compliant but incorrect answer). If several conditions could apply, we keep compilation, runtime, resource, or idle failures as the primary cause; among the remaining rejected runs, protocol violations are labeled before query-budget overruns and incorrect final answers. For discussion, we refer to **WA** as an *algorithmic logic error* and **PE/QLE/IDLE** as *interaction-specific* failures, while **CE/RE/TLE/MLE** capture compilation-, runtime-, and resource-limit failures. We provide representative examples for each category in Appendix D.2.

*Table 2.* pass@1 and pass@5 (higher is better) stratified by difficulty on InteractBench. We sample $n = 10$ solutions per task. Configurations are grouped into reasoning and non-reasoning.

| Model | Easy | | Medium | | Hard | | Overall | |
|---|---|---|---|---|---|---|---|---|
| | **pass@1** | **pass@5** | **pass@1** | **pass@5** | **pass@1** | **pass@5** | **pass@1** | **pass@5** |
| *Reasoning Configurations* | | | | | | | | |
| Gemini-3-Pro-Preview | 0.803 | 0.961 | 0.512 | 0.716 | 0.367 | 0.597 | 0.554 | 0.752 |
| GPT-5.2 | 0.742 | 0.921 | 0.530 | 0.690 | 0.340 | 0.492 | 0.542 | 0.705 |
| DeepSeek-V3.2-Thinking | 0.645 | 0.816 | 0.276 | 0.484 | 0.104 | 0.239 | 0.332 | 0.513 |
| Gemini-2.5-Pro | 0.587 | 0.763 | 0.259 | 0.452 | 0.092 | 0.224 | 0.305 | 0.480 |
| GPT-OSS-120B | 0.521 | 0.750 | 0.199 | 0.355 | 0.078 | 0.179 | 0.254 | 0.416 |
| Claude-Opus-4.5 | 0.318 | 0.592 | 0.141 | 0.284 | 0.021 | 0.060 | 0.159 | 0.312 |
| Gemini-2.5-Flash | 0.332 | 0.566 | 0.144 | 0.290 | 0.009 | 0.045 | 0.162 | 0.305 |
| GPT-OSS-20B | 0.313 | 0.579 | 0.096 | 0.226 | 0.021 | 0.075 | 0.134 | 0.282 |
| Qwen3-30B-A3B-Thinking | 0.190 | 0.408 | 0.104 | 0.213 | 0.012 | 0.060 | 0.105 | 0.228 |
| Qwen3-32B | 0.237 | 0.421 | 0.065 | 0.123 | 0.009 | 0.045 | 0.096 | 0.181 |
| Qwen3-14B | 0.147 | 0.263 | 0.040 | 0.103 | 0.000 | 0.000 | 0.058 | 0.121 |
| DeepSeek-R1-Distill-Llama-70B | 0.105 | 0.224 | 0.035 | 0.077 | 0.000 | 0.000 | 0.045 | 0.097 |
| *Non-Reasoning Configurations* | | | | | | | | |
| DeepSeek-V3.2 | 0.308 | 0.500 | 0.123 | 0.232 | 0.024 | 0.060 | 0.148 | 0.262 |
| GPT-4.1 | 0.163 | 0.342 | 0.039 | 0.090 | 0.003 | 0.015 | 0.062 | 0.138 |
| Qwen3-32B-NonThinking | 0.032 | 0.092 | 0.017 | 0.032 | 0.000 | 0.000 | 0.017 | 0.040 |
| Qwen3-14B-NonThinking | 0.026 | 0.092 | 0.010 | 0.026 | 0.000 | 0.000 | 0.012 | 0.037 |

## 4.3. Experimental Results

**Experiment #1: performance across difficulty tiers.** We begin by examining how interactive performance degrades with problem difficulty. Interactive tasks with unrevealed information at execution time remain far from solved even for frontier reasoning models. We report pass@1 and pass@5 stratified by easy/medium/hard to reveal the scaling curve that can be obscured by a single aggregate score.

Table 2 shows a steep scaling curve with difficulty. We highlight four findings:

- *Hard interactive tasks remain unsolved and strongly discriminative.* With resampling, Easy problems approach saturation for frontier models. For example, Gemini-3-Pro-Preview reaches pass@5 of 0.961. In contrast, Hard problems remain far from solved: the best-performing model reaches 0.597 on hard pass@5. Only two configurations exceed 0.400 on hard pass@5 (Gemini-3-Pro-Preview and GPT-5.2), yielding a clear separation between frontier configurations and the rest of the suite.
- *Gaps between pass@1 and pass@5 quantify the benefit of resampling.* Even on Easy, pass@1 can lag meaningfully behind pass@5, indicating that additional attempts often recover a correct solution by exploring alternative approaches and interaction strategies. In interactive settings, some of this variance also comes from brittle execution details (e.g., formatting, flushing, and protocol adherence), rather than always missing high-level ideas. Appendix D.1 provides a representative same-task case study where a small robustness fix flips a protocol failure into a successful run.
- *Resampling yields uneven gains across models and dif-*

*ficulty tiers.* Some frontier configurations see larger improvements on Hard under resampling, suggesting that correct solutions are often attainable but sensitive to small implementation details. For example, on Hard, Gemini-3-Pro-Preview rises from 0.367 in pass@1 to 0.597 in pass@5, whereas GPT-5.2 rises from 0.340 to 0.492. Others show diminishing gains on Hard, suggesting a more fundamental capability gap.

- *Explicit reasoning improves pass rates, but does not remove the Hard barrier.* Within-family comparisons show large improvements from reasoning variants (e.g., DeepSeek-V3.2-Thinking attains an overall pass@5 of 0.513, compared with 0.262 for DeepSeek-V3.2). However, smaller models remain bottlenecked on Hard even with reasoning (e.g., Qwen3-14B achieves 0.000 hard pass@5), suggesting that explicit reasoning is helpful but insufficient without strong base capability.

**Experiment #2: breakdown by category.** We next localize the interaction gap across categories to identify which categories most stress *information acquisition* and *dynamic state tracking* under unrevealed information.

Table 3 highlights where interactive capability breaks down, and reveals four patterns:

- *Graph is the dominant bottleneck.* Graph is consistently challenging: even the best-performing model stays below 0.500 in pass@1. This reflects the need to acquire information from an initially unrevealed structure under strict query budgets, and to track state under partial observability. In contrast, tasks in the Bit group are comparatively easier across the suite; for example, Gemini-3-Pro-Preview attains a pass@1 of 0.613 on Bit compared with

*Table 3.* pass@1 and pass@5 by category. Configurations are grouped into reasoning and non-reasoning.

| Model | Graph | | Search | | Greedy | | Bit | | Data Structures | | Math | | Game | |
|---|---|---|---|---|---|---|---|---|---|---|---|---|---|---|
| | pass@1 | pass@5 | pass@1 | pass@5 | pass@1 | pass@5 | pass@1 | pass@5 | pass@1 | pass@5 | pass@1 | pass@5 | pass@1 | pass@5 |
| Reasoning Configurations | | | | | | | | | | | | | | |
| Gemini-3-Pro-Preview | 0.475 | 0.705 | 0.561 | 0.770 | 0.470 | 0.725 | 0.613 | 0.711 | 0.487 | 0.750 | 0.589 | 0.744 | 0.480 | 0.743 |
| GPT-5.2 | 0.450 | 0.591 | 0.547 | 0.708 | 0.530 | 0.725 | 0.569 | 0.711 | 0.500 | 0.711 | 0.563 | 0.712 | 0.566 | 0.800 |
| DeepSeek-V3.2-Thinking | 0.295 | 0.511 | 0.340 | 0.504 | 0.345 | 0.580 | 0.338 | 0.467 | 0.295 | 0.513 | 0.314 | 0.472 | 0.377 | 0.657 |
| Gemini-2.5-Pro | 0.255 | 0.443 | 0.311 | 0.469 | 0.293 | 0.478 | 0.342 | 0.444 | 0.247 | 0.447 | 0.307 | 0.472 | 0.309 | 0.600 |
| GPT-OSS-120B | 0.216 | 0.386 | 0.269 | 0.407 | 0.226 | 0.362 | 0.280 | 0.511 | 0.184 | 0.316 | 0.254 | 0.424 | 0.291 | 0.457 |
| Claude-Opus-4.5 | 0.139 | 0.261 | 0.152 | 0.274 | 0.148 | 0.319 | 0.120 | 0.289 | 0.129 | 0.303 | 0.165 | 0.304 | 0.149 | 0.314 |
| Gemini-2.5-Flash | 0.114 | 0.204 | 0.170 | 0.310 | 0.171 | 0.391 | 0.151 | 0.289 | 0.129 | 0.303 | 0.166 | 0.312 | 0.183 | 0.429 |
| GPT-OSS-20B | 0.120 | 0.250 | 0.136 | 0.266 | 0.145 | 0.319 | 0.098 | 0.244 | 0.068 | 0.210 | 0.141 | 0.304 | 0.149 | 0.343 |
| Qwen3-30B-A3B-Thinking | 0.102 | 0.193 | 0.115 | 0.257 | 0.154 | 0.348 | 0.084 | 0.178 | 0.095 | 0.210 | 0.082 | 0.208 | 0.137 | 0.314 |
| Qwen3-32B | 0.111 | 0.182 | 0.101 | 0.195 | 0.113 | 0.232 | 0.098 | 0.156 | 0.074 | 0.145 | 0.056 | 0.128 | 0.120 | 0.229 |
| Qwen3-14B | 0.061 | 0.136 | 0.073 | 0.124 | 0.067 | 0.159 | 0.049 | 0.067 | 0.053 | 0.132 | 0.040 | 0.104 | 0.074 | 0.200 |
| DeepSeek-R1-Distill-Llama-70B | 0.039 | 0.114 | 0.073 | 0.133 | 0.052 | 0.145 | 0.018 | 0.044 | 0.037 | 0.092 | 0.037 | 0.064 | 0.057 | 0.171 |
| Non-Reasoning Configurations | | | | | | | | | | | | | | |
| DeepSeek-V3.2 | 0.139 | 0.261 | 0.165 | 0.283 | 0.159 | 0.290 | 0.138 | 0.244 | 0.095 | 0.237 | 0.131 | 0.216 | 0.131 | 0.229 |
| GPT-4.1 | 0.057 | 0.159 | 0.064 | 0.115 | 0.078 | 0.174 | 0.036 | 0.067 | 0.053 | 0.132 | 0.042 | 0.112 | 0.057 | 0.229 |
| Qwen3-32B-NonThinking | 0.014 | 0.034 | 0.012 | 0.044 | 0.014 | 0.058 | 0.022 | 0.022 | 0.008 | 0.026 | 0.011 | 0.024 | 0.017 | 0.029 |
| Qwen3-14B-NonThinking | 0.021 | 0.045 | 0.012 | 0.053 | 0.015 | 0.043 | 0.000 | 0.000 | 0.011 | 0.040 | 0.005 | 0.016 | 0.017 | 0.057 |

0.475 on Graph. This is consistent with more templated information extraction and less complex dynamic state tracking.

- *Frontier models exhibit complementary strengths across categories.* Gemini-3-Pro-Preview is especially strong on Graph, whereas GPT-5.2 is strongest on Game, suggesting that interactive performance decomposes into distinct skills and that aggregate scores can hide specialization.
- *Stateful maintenance remains a recurring source of difficulty.* Data Structures tasks in our taxonomy often behave like online reconstruction and maintenance: the solver must maintain a structured internal state (e.g., refining an order/permutation/partition from relational tests) and update it correctly after each reply while preserving invariants. This long-horizon bookkeeping amplifies small update mistakes over many rounds.
- *Reasoning lifts performance broadly across categories.* Within-family comparisons show consistent gains across columns; for example, DeepSeek-V3.2-Thinking improves over DeepSeek-V3.2 on Graph, raising pass@5 from 0.261 to 0.511, with similar improvements on Search, Greedy, Math, and Game.

**Experiment #3: failure-mode decomposition.** Finally, we diagnose failure causes: whether the bottleneck is algorithmic logic errors (WA) or interaction-specific breakdowns. We decompose failed executions using our interaction-aware taxonomy (Section 4.2) and report the fraction attributable to each primary failure type.

Table 4 yields three insights into failure causes in interactive runs:

- *Algorithmic logic errors (WA) dominate, but interaction-specific failures occur at different stages.* In Table 4, **WA** is the most common failure for many configurations, indicating that interactive tasks remain bottlenecked by algorithmic reasoning and state tracking under partial observability. The interaction-specific labels further dis-

*Table 4.* Failure composition among unsuccessful executions (higher means more common). We condition on failed runs and report the fraction of failures attributable to each primary failure type for each model (i.e., normalized by that model's failed runs; Section 4.2). Configurations are grouped into reasoning and non-reasoning.

| Model | IDLE | PE | CE | TLE | MLE | RE | WA | QLE |
|---|---|---|---|---|---|---|---|---|
| Reasoning Configurations | | | | | | | | |
| Gemini-3-Pro-Preview | 0.018 | 0.120 | 0.027 | 0.078 | 0.003 | 0.011 | 0.624 | 0.119 |
| GPT-5.2 | 0.019 | 0.158 | 0.031 | 0.064 | 0.000 | 0.015 | 0.612 | 0.101 |
| DeepSeek-V3.2-Thinking | 0.024 | 0.132 | 0.063 | 0.045 | 0.002 | 0.012 | 0.647 | 0.074 |
| Gemini-2.5-Pro | 0.046 | 0.157 | 0.024 | 0.035 | 0.002 | 0.011 | 0.658 | 0.068 |
| GPT-OSS-120B | 0.401 | 0.105 | 0.013 | 0.013 | 0.002 | 0.010 | 0.424 | 0.031 |
| Claude-Opus-4.5 | 0.012 | 0.124 | 0.201 | 0.067 | 0.006 | 0.003 | 0.505 | 0.082 |
| Gemini-2.5-Flash | 0.035 | 0.141 | 0.041 | 0.028 | 0.004 | 0.013 | 0.705 | 0.033 |
| GPT-OSS-20B | 0.094 | 0.125 | 0.074 | 0.016 | 0.002 | 0.009 | 0.653 | 0.027 |
| Qwen3-30B-A3B-Thinking | 0.051 | 0.118 | 0.023 | 0.020 | 0.002 | 0.020 | 0.730 | 0.036 |
| Qwen3-32B | 0.039 | 0.160 | 0.034 | 0.028 | 0.001 | 0.011 | 0.683 | 0.044 |
| Qwen3-14B | 0.052 | 0.149 | 0.047 | 0.024 | 0.001 | 0.014 | 0.686 | 0.028 |
| DeepSeek-R1-Distill-Llama-70B | 0.071 | 0.188 | 0.106 | 0.025 | 0.002 | 0.015 | 0.564 | 0.028 |
| Non-Reasoning Configurations | | | | | | | | |
| DeepSeek-V3.2 | 0.040 | 0.171 | 0.059 | 0.039 | 0.004 | 0.011 | 0.612 | 0.063 |
| GPT-4.1 | 0.034 | 0.200 | 0.069 | 0.042 | 0.002 | 0.010 | 0.579 | 0.064 |
| Qwen3-32B-NonThinking | 0.048 | 0.137 | 0.121 | 0.023 | 0.001 | 0.014 | 0.623 | 0.035 |
| Qwen3-14B-NonThinking | 0.079 | 0.187 | 0.139 | 0.031 | 0.002 | 0.021 | 0.524 | 0.018 |

tinguish where execution breaks: **PE** and **IDLE** usually indicate that execution breaks before a stable query loop is established, whereas **QLE** indicates that the solver enters the protocol but exhausts the query budget. Some configurations still break down at the earlier stage, notably elevated **IDLE** for GPT-OSS-120B; qualitative cases suggest it may misinterpret interactive tasks as ordinary input-output problems and never enter a valid query loop. Appendix D.1 provides representative cases.

- *Query efficiency limits frontier models as well.* **QLE** is more prevalent among frontier models (e.g., 0.119 for Gemini-3-Pro-Preview and 0.101 for GPT-5.2) than among weaker ones, suggesting that stronger models can follow the protocol but still fail to manage the query budget.
- *Runtime and resource-limit failures are present but secondary.* **TLE**, **MLE**, and **RE** appear across the suite, but are generally secondary to algorithmic logic errors and interaction-specific failures.

We provide representative qualitative examples for each failure type in Appendix D.2. Appendix C.1 complements this analysis with a batch-vs-interactive comparison: revealing the hidden test data improves pass@1 and reduces protocol errors and query-budget overruns, while wrong answers remain the main residual failure type. Appendix C.2 then compares independent resampling, few-shot prompting, and iterative refinement. Few-shot prompting and independent resampling perform similarly overall, whereas iterative refinement is more capability-dependent and mainly repairs execution-level failures.

## 5. Conclusion

InteractBench targets competitive-programming settings where key information is *unrevealed at execution time* and must be acquired through multi-round queries under strict protocols and limited query budgets. We curate 322 interactive tasks from Codeforces, AtCoder, IOI, and ICPC, and package each task with an executable local interactor, enabling fully offline and reproducible evaluation. Beyond pass@k, our harness surfaces interaction-specific failure modes via protocol compliance checks, query-budget enforcement, and a fine-grained taxonomy that separates algorithmic errors from interaction breakdowns.

Across 16 model configurations, we observe consistent degradation as problem difficulty increases, highlighting a clear gap between standard (full-information) coding benchmarks and interactive problem solving. While failures are dominated by algorithmic logic errors, protocol violations and query-budget overruns remain frequent even for frontier reasoning models.

Looking forward, we plan to expand coverage and maintain freshness through versioned releases, and to develop prompting and training strategies that directly target interactive querying, state maintenance, and protocol adherence.

## 6. Limitations

Scale remains the main practical limitation of InteractBench. Interactive tasks are much less common than batch-style problems on competitive-programming platforms, and offline evaluation requires a reliable local interactor. Future versions will expand coverage by collecting more eligible tasks and constructing new ones with verified interactors. In addition, our experiments use zero-shot model-generated solvers, and follow-up studies can examine prompting and training strategies for interactive querying, state maintenance, and protocol adherence.

## Impact Statement

InteractBench is a benchmark for evaluating program synthesis under unrevealed information, where a generated solver must follow a strict interaction protocol and acquire information through multi-round queries. We expect it to support research on reliable code generation, state tracking, and protocol adherence, and to enable fine-grained, reproducible evaluation of LLM-based coding systems. More broadly, this paper presents work whose goal is to advance the field of machine learning by providing an evaluation methodology for model capabilities. We encourage responsible use in accordance with applicable rules and norms.

## Acknowledgments

Qiankun Zhang is supported by the National Natural Science Foundation of China (Grant 62302183), Open Foundation of Key Laboratory of Cyberspace Security, Ministry of Education of China (Grant KLCS20240401), and CCF-DiDi GAIA Collaborative Research Funds (Grant CCF-DiDi GAIA 202522). Xianjun Deng is supported by the National Key R&D Program of China (Grant 2022YFE0138600), and the National Natural Science Foundation of China (Grant U24B20153).

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

# Appendix

# A. InteractBench Specification

This appendix collects benchmark specifications and supplementary materials for InteractBench. We organize the appendix into four parts: (i) specification, (ii) evaluation setup, (iii) supplementary experiments, and (iv) diagnostics and artifacts.

## A.1. Task Card

Each InteractBench task is accompanied by a structured task card specifying the task interface required by our offline harness. We include the schema here to make the benchmark interface explicit and auditable.

*Table 5.* Task card fields in InteractBench.

| Field | Description |
| --- | --- |
| problem_id | Unique identifier of the task. |
| desc | Problem statement, including the stdin/stdout interaction protocol, termination rules, and constraints (e.g., query limits). |
| interactor_mode | Interactor mode metadata: non-adaptive, adaptive, or both. |
| cpu_time_limit_ms | CPU time limit (in milliseconds). |
| memory_limit_mb | Memory limit (in megabytes). |
| test_cases | Hidden test cases used for offline evaluation and consumed by the interactor. |
| interactor | Interactor program that implements the protocol and checks the solver's behavior. |
| generator | Optional generator program used to produce hidden test cases. |
| difficulty | Task difficulty label. |
| categorization | Category labels assigned to the task (e.g., Graph, Search). |

# B. Evaluation Setup

## B.1. Dataset Statistics

Experiments #1–2 in Section 4 use InteractBench. Section 3 describes how difficulty and category labels are assigned. This appendix reports the source, difficulty, and category breakdowns and gives the full category definitions.

**Task sources and difficulty.** InteractBench tasks are curated from four competitive-programming sources: Codeforces (Codeforces, 2026) and AtCoder (AtCoder Inc., 2026) (online judges), and the IOI (International Olympiad in Informatics, 2026) and ICPC (ICPC Foundation, 2026) series (competition series). Table 6 reports task counts by source and difficulty.

**Category breakdown.** Table 7 reports task counts by category stratified by difficulty. Because category labels are multi-label, a task may contribute to more than one category.

**Category definitions.** For label assignment and category-level analyses, we use the definitions below.

- **Graph.** The hidden object is naturally modeled as a graph or tree, or a hidden target is defined on a known graph structure. Queries usually reveal relational or metric-like signals, such as connectivity, reachability, distances, neighborhoods, or objective values on graph elements. Solvers choose probes from the partial structure inferred so far.
- **Search.** The solver localizes a hidden value, position, or configuration by shrinking a feasible set. Replies provide constraints such as membership, counts over a subset or region, monotone comparisons, or directional feedback. The solution maintains the feasible region across rounds and handles boundary cases at termination.
- **Greedy.** The solver builds an answer through committed steps whose feedback constrains later moves. The key invariant is that the partial construction remains extendable; otherwise locally plausible decisions can lead to dead ends or waste interaction opportunities.

- **Bit.** Replies carry low-bandwidth information, often Boolean or parity-like. Solvers either design queries that collect independent constraints or decode a short transcript into the hidden object under the protocol restrictions.
- **Data Structures.** These tasks require online maintenance of structured state across replies. Typical state includes orders, permutations, partitions, or reconstruction invariants, where bookkeeping errors affect later queries and the final answer.
- **Math.** Replies follow arithmetic, algebraic, or geometric structure that supports deduction or constructive design from a small number of queries. The solver combines these relations to identify an unknown object or construct a valid witness, while handling uniqueness and degenerate cases.
- **Game.** The solver must remain correct against worst-case replies allowed by the protocol. Solutions reason from what the transcript guarantees and often use minimax-style planning over the remaining knowledge states.

*Table 6.* Task counts by difficulty, broken down by source.

| Source | Easy | Medium | Hard | Total |
|---|---|---|---|---|
| Codeforces | 52 | 115 | 53 | 220 |
| ICPC | 16 | 33 | 12 | 61 |
| AtCoder | 8 | 7 | 2 | 17 |
| IOI | 9 | 7 | 8 | 24 |
| **Total** | **85** | **162** | **75** | **322** |

*Table 7.* Task counts by category stratified by difficulty. A task may be assigned multiple categories.

| Category | Easy | Medium | Hard | Total |
|---|---|---|---|---|
| Graph | 23 | 45 | 20 | 88 |
| Search | 31 | 58 | 24 | 113 |
| Greedy | 16 | 42 | 11 | 69 |
| Bit | 12 | 22 | 11 | 45 |
| Data Structures | 14 | 51 | 11 | 76 |
| Math | 31 | 60 | 34 | 125 |
| Game | 11 | 18 | 6 | 35 |

## B.2. Model Card

*Table 8.* **Model card for models used in InteractBench.** Models are grouped by provider and access mode (closed-source or open-source). We list parameter counts when publicly specified and provide official hyperlinks.

| Model | Mode | Size | Hyperlink |
|---|---|---|---|
| *OpenAI (Closed-source)* | | | |
| GPT-5.2 | Thinking | – | https://chatgpt.com |
| GPT-4.1 | NonThinking | – | https://chatgpt.com |
| *OpenAI (Open-source)* | | | |
| GPT-OSS-120B | Thinking | 120B | https://huggingface.co/openai/gpt-oss-120b |
| GPT-OSS-20B | Thinking | 20B | https://huggingface.co/openai/gpt-oss-20b |
| *Google (Closed-source)* | | | |
| Gemini-3-Pro-Preview | Thinking | – | https://gemini.google.com |
| Gemini-2.5-Pro | Thinking | – | https://gemini.google.com |
| Gemini-2.5-Flash | Thinking | – | https://gemini.google.com |
| *Anthropic (Closed-source)* | | | |
| Claude-Opus-4.5 | Thinking | – | https://claude.ai |
| *DeepSeek (Open-source)* | | | |
| DeepSeek-V3.2-Thinking | Thinking | 685B | https://huggingface.co/deepseek-ai/DeepSeek-V3.2 |
| DeepSeek-V3.2 | NonThinking | 685B | https://huggingface.co/deepseek-ai/DeepSeek-V3.2 |
| DeepSeek-R1-Distill-Llama-70B | Thinking | 70B | https://huggingface.co/deepseek-ai/DeepSeek-R1-Distill-Llama-70B |
| *Qwen/Alibaba (Open-source)* | | | |
| Qwen3-32B | Thinking | 32B | https://huggingface.co/Qwen/Qwen3-32B |
| Qwen3-32B-NonThinking | NonThinking | 32B | https://huggingface.co/Qwen/Qwen3-32B |
| Qwen3-30B-A3B-Thinking | Thinking | 30B-A3B | https://huggingface.co/Qwen/Qwen3-30B-A3B-Thinking-2507 |
| Qwen3-14B | Thinking | 14B | https://huggingface.co/Qwen/Qwen3-14B |
| Qwen3-14B-NonThinking | NonThinking | 14B | https://huggingface.co/Qwen/Qwen3-14B |

## B.3. Evaluation Prompt for Code Generation

We evaluate models with a standardized zero-shot wrapper prompt. The wrapper is shown as a language-parameterized template (placeholders such as {LANGUAGE} / {LANGUAGE_TAG}) and instantiated with the target language in each experiment. The wrapper is intentionally minimal to reduce prompt-engineering confounds, while still enforcing interactive I/O hygiene (protocol compliance, flushing, and no extraneous output). Models are evaluated by compiling and executing the generated program against the task interactor; correctness is determined by the interactive transcript under the protocol.

---

**Evaluation prompt: code generation**

```
You are a competitive programming expert solving an INTERACTIVE problem.
Implement a complete {LANGUAGE} solution that deduces hidden parameters strictly through the defined query
    protocol.
Your solution will be compiled and executed against an interactive judge; only the program's interactive
    behavior under the protocol is evaluated.

Requirements:
- Single-file program with a standard entry point (e.g., main / top-level), reading from stdin and writing to
    stdout.
- Flush stdout after every query using the idiomatic mechanism in {LANGUAGE} (examples):
  - C++: 'std::endl', 'std::cout << std::flush', or 'std::cout.flush()'.
  - Python: 'print(..., flush=True)' or 'sys.stdout.flush()'.
  - Java: 'System.out.flush()'.
  - Go: if using 'bufio.Writer', call 'w.Flush()' after each query.
  - Rust: 'use std::io::Write; std::io::stdout().flush().unwrap();'.
- Terminate immediately if the judge returns an invalid response (e.g., -1).
- No debug output to stdout.

Output format:
- Output a single fenced code block containing the complete program (use the correct language identifier, e.g.,
    cpp/python/java), and nothing else:
'''{LANGUAGE_TAG}
// code...
'''
```

---

# C. Supplementary Experiments

This section collects supplementary experiments that complement the main experiments in Section 4.

## C.1. Batch vs. Interactive Evaluation

We compare a batch variant against the standard interactive evaluation reported in Section 4 on 150 matched problems. In the batch variant, the solver receives the hidden test data upfront through standard input and still has to produce the final answer under the same validation logic. This keeps the final validation unchanged while removing the need to acquire hidden information through queries.

*Table 9.* **Batch vs. interactive evaluation.** pass@1 and failure composition on 150 matched problems. Failure-type fractions are computed over unsuccessful executions.

| Model | Setting | pass@1 | WA | PE | QLE | IDLE | CE | RE | TLE | MLE |
|---|---|---|---|---|---|---|---|---|---|---|
| GPT-5.2 | Batch | 0.767 | 0.800 | 0.086 | 0.000 | 0.029 | 0.029 | 0.000 | 0.057 | 0.000 |
| | Interactive | 0.511 | 0.629 | 0.164 | 0.076 | 0.005 | 0.038 | 0.008 | 0.079 | 0.000 |
| DeepSeek-V3.2-Thinking | Batch | 0.560 | 0.788 | 0.106 | 0.015 | 0.015 | 0.030 | 0.015 | 0.030 | 0.000 |
| | Interactive | 0.313 | 0.692 | 0.115 | 0.059 | 0.014 | 0.060 | 0.010 | 0.047 | 0.004 |
| DeepSeek-V3.2 | Batch | 0.367 | 0.695 | 0.084 | 0.032 | 0.021 | 0.137 | 0.011 | 0.021 | 0.000 |
| | Interactive | 0.147 | 0.653 | 0.178 | 0.033 | 0.028 | 0.041 | 0.017 | 0.046 | 0.003 |
| Qwen3-14B | Batch | 0.093 | 0.794 | 0.088 | 0.022 | 0.015 | 0.051 | 0.015 | 0.015 | 0.000 |
| | Interactive | 0.056 | 0.681 | 0.150 | 0.033 | 0.052 | 0.041 | 0.017 | 0.024 | 0.003 |
| Qwen3-14B-NonThinking | Batch | 0.180 | 0.691 | 0.114 | 0.016 | 0.024 | 0.130 | 0.008 | 0.016 | 0.000 |
| | Interactive | 0.011 | 0.535 | 0.190 | 0.014 | 0.096 | 0.115 | 0.023 | 0.027 | 0.001 |

Table 9 shows that revealing the hidden test data improves pass@1 for every tested configuration. The gain is largest for GPT-5.2, DeepSeek-V3.2-Thinking, and DeepSeek-V3.2, while Qwen3-14B shows a smaller improvement. A rank reversal appears for the two Qwen3-14B variants: Qwen3-14B-NonThinking is higher in batch evaluation (0.180 vs. 0.093), but lower in interactive evaluation (0.011 vs. 0.056). Batch evaluation also substantially reduces protocol errors and query-budget overruns, but wrong answers remain the dominant failure type among unsuccessful runs. Overall, unrevealed information adds a distinct source of difficulty beyond the batch algorithmic core. At the same time, the remaining wrong answers indicate that revealing the hidden data upfront does not eliminate the underlying reasoning problem.

### C.2. Resampling, Few-Shot Prompting, and Iterative Refinement

We further compare three evaluation-time strategies: independent resampling, few-shot prompting with a solved interactive example, and iterative refinement. Following ICPC-Eval (Xu et al., 2025), iterative refinement runs the solver for several rounds: after each failed attempt, the execution feedback is returned to the model, which then revises its program for the next round. refine@k is the cumulative solve rate within $k$ refinement rounds.

*Table 10.* **Comparison of mitigation strategies.** Independent resampling (pass@5), few-shot prompting, and iterative refinement (refine@5).

| Model | pass@5 | pass@5 (few-shot) | refine@5 |
|---|---|---|---|
| GPT-5.2 | 0.705 | 0.714 | 0.677 |
| DeepSeek-V3.2-Thinking | 0.513 | 0.506 | 0.491 |
| DeepSeek-V3.2 | 0.262 | 0.279 | 0.227 |
| Qwen3-14B | 0.121 | 0.143 | 0.071 |
| Qwen3-14B-NonThinking | 0.037 | 0.056 | 0.019 |

Table 10 shows that few-shot prompting is competitive with independent resampling and is slightly higher for most configurations. The clearest few-shot gains appear for weaker configurations, such as Qwen3-14B and Qwen3-14B-NonThinking. Iterative refinement improves over a single attempt for stronger models, but its final scores remain below those of independent resampling and few-shot prompting.

*Table 11.* **Per-round failure composition under iterative refinement.** Error-type fractions among still-unsolved executions at rounds 1, 3, and 5, together with the cumulative solve rate refine@k, across five representative model configurations.

| Model | Round | CE | PE | IDLE | WA | QLE | TLE | MLE | RE | refine@k |
|---|---|---|---|---|---|---|---|---|---|---|
| GPT-5.2 | r1 | 0.040 | 0.139 | 0.020 | 0.609 | 0.113 | 0.060 | 0.007 | 0.013 | 0.531 |
| | r3 | 0.016 | 0.107 | 0.008 | 0.590 | 0.115 | 0.049 | 0.000 | 0.115 | 0.621 |
| | r5 | 0.010 | 0.038 | 0.010 | 0.625 | 0.135 | 0.048 | 0.000 | 0.135 | 0.677 |
| DeepSeek-V3.2-Thinking | r1 | 0.087 | 0.097 | 0.015 | 0.655 | 0.092 | 0.039 | 0.000 | 0.015 | 0.360 |
| | r3 | 0.034 | 0.069 | 0.006 | 0.632 | 0.098 | 0.034 | 0.006 | 0.121 | 0.460 |
| | r5 | 0.024 | 0.030 | 0.006 | 0.646 | 0.098 | 0.030 | 0.006 | 0.159 | 0.491 |
| DeepSeek-V3.2 | r1 | 0.054 | 0.193 | 0.032 | 0.564 | 0.100 | 0.046 | 0.000 | 0.011 | 0.130 |
| | r3 | 0.020 | 0.141 | 0.023 | 0.574 | 0.102 | 0.039 | 0.000 | 0.102 | 0.205 |
| | r5 | 0.008 | 0.124 | 0.012 | 0.578 | 0.100 | 0.028 | 0.004 | 0.145 | 0.227 |
| Qwen3-14B | r1 | 0.059 | 0.147 | 0.049 | 0.657 | 0.062 | 0.016 | 0.000 | 0.010 | 0.050 |
| | r3 | 0.030 | 0.107 | 0.033 | 0.673 | 0.063 | 0.013 | 0.003 | 0.077 | 0.068 |
| | r5 | 0.023 | 0.090 | 0.017 | 0.672 | 0.064 | 0.010 | 0.000 | 0.124 | 0.071 |
| Qwen3-14B-NonThinking | r1 | 0.144 | 0.172 | 0.110 | 0.502 | 0.028 | 0.028 | 0.000 | 0.016 | 0.009 |
| | r3 | 0.063 | 0.142 | 0.069 | 0.571 | 0.032 | 0.025 | 0.000 | 0.098 | 0.016 |
| | r5 | 0.025 | 0.111 | 0.041 | 0.589 | 0.032 | 0.019 | 0.003 | 0.180 | 0.019 |

Table 11 further shows that refinement can reduce execution-level failures, especially compilation and protocol errors. For GPT-5.2, for example, CE decreases from 0.040 at round 1 to 0.010 at round 5, and PE decreases from 0.139 to 0.038. At the same time, WA remains the dominant residual failure type after refinement for all five configurations. This pattern suggests that feedback can repair some execution-level issues, but many remaining failures still require solving the underlying

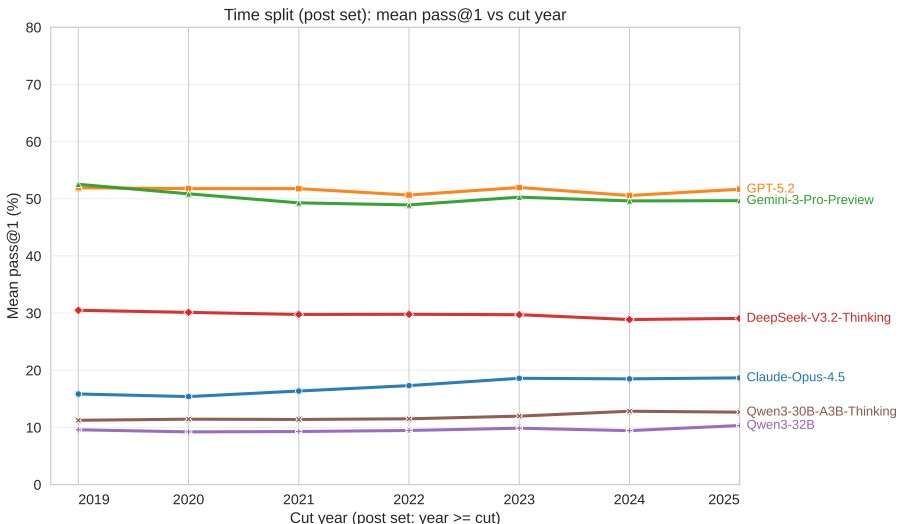

*Figure 3.* **Temporal contamination diagnostic via time splits.** For each cut year $Y$, we compute the mean pass@1 over problems released in years $\geq Y$ (the post set).

partially observed interactive reasoning problem.

### C.3. Temporal Contamination Diagnostic via Time Splits

We use a time-split diagnostic on InteractBench to probe potential temporal data contamination for tasks with known release years. For a cut year $Y$, we define a *pre* set $P_{<Y}$ (problems released in years $< Y$) and a *post* set $P_{\geq Y}$ (problems released in years $\geq Y$), and report the mean pass@1 on $P_{\geq Y}$ as a function of $Y$. Intuitively, if memorization or leakage concentrated around a particular time window dominates performance, we may observe a sharp change (a "cliff") as the cut moves past that window; smoother curves suggest that no single sharp temporal boundary drives the score.

**Results.** Figure 3 shows the post-set mean pass@1 curves for six representative model configurations spanning major providers and open weights: Claude-Opus-4.5, GPT-5.2, Gemini-3-Pro-Preview, DeepSeek-V3.2-Thinking, Qwen3-32B, and Qwen3-30B-A3B-Thinking. Across the sweep, we do not observe a cliff-like discontinuity as the cut year advances; instead, the curves change gradually for all models.

### C.4. Multi-language and I/O Template Evaluation

We evaluate five representative model configurations on InteractBench across C++, Python, Java, and Go under the same packaged local interactors. For each language, we compare the standard zero-shot prompt with a prompt that includes the interactive I/O template in Appendix D.7. To account for language runtime overhead, we relax time limits for Python and Java. Because the main experiments use C++ without templates, this section reports whether the same model ordering appears under other language and template settings.

*Table 12.* **Multi-language and I/O template evaluation.** pass@1 on interactive tasks executable across C++/Python/Java/Go.

| Model | Setting | C++ | Python | Java | Go |
|---|---|---|---|---|---|
| GPT-5.2 | no template | 0.542 | 0.550 | 0.564 | 0.550 |
| | I/O template | 0.557 | 0.544 | 0.544 | 0.540 |
| DeepSeek-V3.2-Thinking | no template | 0.332 | 0.309 | 0.312 | 0.242 |
| | I/O template | 0.319 | 0.289 | 0.285 | 0.285 |
| DeepSeek-V3.2 | no template | 0.148 | 0.060 | 0.134 | 0.104 |
| | I/O template | 0.144 | 0.131 | 0.107 | 0.124 |
| Qwen3-14B | no template | 0.058 | 0.060 | 0.067 | 0.027 |
| | I/O template | 0.060 | 0.067 | 0.077 | 0.013 |
| Qwen3-14B-NonThinking | no template | 0.012 | 0.007 | 0.010 | 0.000 |
| | I/O template | 0.017 | 0.013 | 0.013 | 0.003 |

Table 12 reports pass@1 for each configuration.

The template effect is mixed rather than uniformly beneficial. For DeepSeek-V3.2 on Python, adding the template improves pass@1 from 0.060 to 0.131; for DeepSeek-V3.2-Thinking on Python, it slightly decreases pass@1 from 0.309 to 0.289. This pattern is consistent with the failure labels. Python without templates more often leads to idle or deadlocked runs, so the template mainly reduces low-level I/O failures rather than resolving the algorithmic reasoning errors. Java and Go introduce more compile failures, especially for weaker models. After these language-specific failures are separated, wrong answers remain the main source of unsuccessful runs, consistent with the failure analysis in Section 4. C++ shows the smallest difference between template and no-template settings, supporting our use of C++ without templates as the default evaluation configuration.

## D. Diagnostics and Artifacts

### D.1. Qualitative Evidence for Key Findings

Section 4 reports aggregate pass rates and failure compositions for interactive problem solving. To make two of the most salient phenomena concrete, we manually inspected a small set of representative tasks and model-generated submissions. The examples below are *illustrative rather than exhaustive*: they highlight (i) how resampling can "rescue" runs by exploring alternative approaches and avoiding small robustness bugs, and (ii) how some models exhibit disproportionately frequent interaction-level breakdowns that prevent meaningful progress (notably **IDLE**).

---

**Resampling can recover from protocol/implementation brittleness**

**Observation.** In Experiment #1, pass@1 can lag noticeably behind pass@5 even on easier tasks, suggesting that some failures are caused not by missing high-level ideas, but by small protocol or robustness issues that resampling may "rescue."

**Source task.** cf2066A. **Model.** DeepSeek-V3.2.

**Harness evidence (failing sample).** The submission fails immediately with a format error before issuing any valid query:

```
verdict=PE
wrong output format Invalid query indices
queries=0  query_limit=2
```

**Key code difference.** In the failing sample, the solver attempts to query an index pair that may remain uninitialized:

```
int p = -1, q = -1;
...
cout << "? " << p << " " << q << endl;
```

A second sample for the same model follows the same high-level strategy but adds a simple fallback that guarantees valid indices before querying:

```
if (a == -1) { a = 1; b = 2; }  // fallback to a valid pair
```

---

> **Takeaway.** This pair illustrates a common "recoverable" pattern in interactive evaluation: a plausible strategy is present, but brittle edge handling causes a protocol failure. Resampling can help by producing a nearby implementation that respects the protocol.

> **GPT-OSS-120B: frequent IDLE breakdowns**
>
> **Observation.** In Experiment #3, GPT-OSS-120B shows an unusually high share of **IDLE** failures. Many of these stalls occur without issuing any query, consistent with raw generations in which the model misinterprets interactive tasks as ordinary offline input-output settings and blocks waiting for hidden inputs. This contrasts with frontier models that more often execute the protocol correctly and fail as **WA**.
>
> **Case 1 (queries=0 IDLE from offline misinterpretation).** On cf2181C, the raw generation explicitly assumes an offline version and reads an edge list, rather than issuing interactive queries. The harness reports:
>
> ```
> verdict=IDLE
> queries=0   query_limit=8
> ```
>
> The saved `.cpp` begins by reading `m` edges:
>
> ```cpp
> int m;
> cin >> m;
> for (int i = 1; i <= m; ++i) {
>     int u, v;
>     cin >> u >> v;
>     ...
> }
> ```
>
> Under the true interactive protocol, no such edge list is ever provided, so the solver blocks immediately and is classified as **IDLE**.
>
> **Case 2 (late IDLE from brittle sentinel handling).** On cf1979F, a submission issues thousands of queries but stalls near the end of interaction. The harness reports:
>
> ```
> verdict=IDLE
> queries=2868
> ```
>
> One plausible mechanism suggested by the saved code is brittle error handling around the *termination sentinel*. In this protocol, an invalid query or budget violation may cause the judge to return a single −1 token, after which the solver should terminate immediately. However, the submission unconditionally reads *two* integers before checking whether the first one is −1:
>
> ```cpp
> cout << "? " << d << endl;
> cout.flush();
> int v, u;
> cin >> v >> u;
> if (v == -1) return 0;
> ```
>
> If the judge emits only −1 (with no second integer), the second read can block, producing a "late" **IDLE** after many otherwise protocol-shaped queries.
>
> **Takeaway.** These examples illustrate why GPT-OSS-120B's failure composition can differ qualitatively from models that reliably emit well-formed submissions and reach the **WA** stage: in interactive evaluation, a model may fail to enter a valid query loop at all (e.g., by assuming an ordinary offline setting), and output discipline and I/O protocol robustness can become the dominant bottleneck.

## D.2. Error Case Studies

We provide one representative example for each error type in our interaction-aware failure taxonomy. Each example summarizes the interaction protocol, includes the relevant excerpt from the model-generated submission, and reports the key harness evidence together with a short diagnosis of the underlying failure mode.

**Legend.** We use the following abbreviations: CE (compile error), PE (protocol/format error), IDLE (deadlock/idle timeout), QLE (query-budget overrun), RE (runtime error; often stream desynchronization), TLE (time limit exceeded), MLE

(memory limit exceeded), and `WA` (wrong answer under a protocol-correct run).

---

**CE: compile error caused by flushing the input stream**

**Source task.** 2025 ICPC Central Europe Regional Contest. **Model.** Qwen3-32B.
**Problem and protocol (paraphrased).**
- Hidden vector $A$ of length $N$ (fixed before interaction; $N \leq 60$).
- Query: print ? b1 b2 ... bN, then read an integer reply.
- Final answer: print ! a1 a2 ... aN and terminate.
- Flushing is required after each output line.

**Full statement (source). Interactive note.** This task is interactive. After printing each line, flush the output buffer. For example: `cout << flush` or `cout.flush()` in C++, `System.out.flush()` in Java, and `sys.stdout.flush()` in Python.

**Problem.** There is a hidden sequence $A = (a_1, \ldots, a_N)$ of length $N$, where each $a_i$ is an integer in the range $0 \leq a_i < 2^N$. Your task is to recover $A$ using queries.

In each query you provide a sequence $B = (b_1, \ldots, b_N)$ with $0 \leq b_i < 2^N$. The reply is computed as follows:
- Define $C = (c_1, \ldots, c_N)$ where $c_i = a_i \oplus b_i$ and $\oplus$ denotes bitwise XOR.
- Let $S$ be the set of values obtainable by XOR-ing an arbitrary subset of $\{c_1, \ldots, c_N\}$, including the empty subset whose XOR is 0.
- The answer to the query is $|S|$.

**Example.** If $A = (1, 4, 3)$ and $B = (0, 4, 7)$ then $C = (1 \oplus 0,\ 4 \oplus 4,\ 3 \oplus 7) = (1, 0, 4)$ and $S = \{0, 1, 4, 5\}$, hence the answer is $4$.

**XOR reminder.** The bitwise XOR of two numbers $x$ and $y$ has bit $i$ set iff exactly one of $x$ and $y$ has bit $i$ set. For example, $5 \oplus 3 = 6$ since $101_2 \oplus 011_2 = 110_2$. In C++ and Python, XOR is written as `^`.

**Interaction protocol.** The sequence $A$ is fixed at the start and does not depend on the queries.
- Read an integer $N$ with $1 \leq N \leq 60$.
- To ask a query, print ? b1 b2 ... bN, flush, and read one integer reply.
- When ready, print ! a1 a2 ... aN, flush, and terminate immediately.

**Local testing.** The original statement provides a local grader for testing; its behavior may differ from the official interactor.

**Incorrect code excerpt (model-generated C++).**

```
int N;
cin >> N;
cin.flush();  // BUG: std::istream has no flush()
```

**Harness evidence.** The compiler rejects the submission:

```
error: 'std::istream' has no member named 'flush'
    cin.flush();
        ^~~~~
```

**Why it fails.** Interactive statements emphasize flushing after *printing* queries, but the submission attempts to flush the input stream. InteractBench classifies this as `CE` to separate compilation failures from interaction-logic errors.

---

**PE: protocol violation by emitting a query during the answer phase**

**Source task.** 2025 ICPC Asia Yokohama Regional Contest. **Model.** Gemini-2.5-Pro.
**Problem and protocol (paraphrased).**
- Input provides $n$ stars and their squared distances to the origin.
- Query up to a budget ($\leq 300$): print measure i j and read the squared distance between stars $i$ and $j$.
- After finishing measurements: print a line answer, then print the upper-triangular matrix of all pairwise squared distances.
- Once answer starts, no further measure commands are allowed.

**Full statement (source). Problem.** The positions of $n$ stars are lattice points in $\mathbb{Z}^3$ under a Cartesian coordinate system whose origin is the Earth. You are given the squared distance from the origin to each star, but the directions are

unknown. In one measurement you specify two distinct stars, and the instrument returns their squared distance. Your task is to determine the squared distances between all pairs of stars within a limited number of measurements.

**Interaction.**
- Read an integer $n$ ($2 \leq n \leq 100$). The stars are numbered $1, \ldots, n$.
- Read $n$ integers; the $i$-th is the squared distance from the origin to star $i$. It is guaranteed that all star coordinates are integers in $[-4000, 4000]$, no two stars coincide, and no star is at the origin.
- You may perform up to $300$ measurements. To measure stars $i \neq j$, output `measure i j`, flush, and read one integer reply $d_{i,j}$, the squared distance between stars $i$ and $j$.
- After determining all pairwise distances, output a line containing only `answer`, followed by $n - 1$ lines that list the upper-triangular distances:
  - Line 1: $d_{1,2}$ $d_{1,3}$ $\ldots$ $d_{1,n}$
  - Line 2: $d_{2,3}$ $\ldots$ $d_{2,n}$
  - $\ldots$
  - Line $n - 1$: $d_{n-1,n}$
- After printing the answer block, terminate without extra output.

**Notes.** If your output format is invalid, or if you exceed $300$ measurements, the verdict is Wrong Answer. The coordinates of the stars are fixed before the interaction starts and do not change during the interaction. The original statement provides sample interactions, figures, and a local testing tool.

**Incorrect code excerpt (model-generated C++).**

```
long long ask_dist_sq(int i, int j) {
  cout << "measure " << i << " " << j << endl;
  long long d_sq;
  cin >> d_sq;
  return d_sq;
}

if (N == 2) {
  cout << "answer" << endl;
  cout << ask_dist_sq(1, 2) << endl;   // BUG: emits "measure" during answer phase
  return 0;
}
```

**Harness evidence.** The interactor reports a format error while parsing the numeric answer payload:

```
wrong output format Invalid number in answer: 'measure'.
```

**Why it fails.** The submission mixes protocol phases. It begins the final-answer block, but then issues a new query. This is diagnosed as PE because the output stream no longer matches the grammar expected by the interactor at that stage.

---

**IDLE: deadlock from missing flush after printing the final answer**

**Source task.** 2025 ICPC Asia Pacific Championship. **Model.** GPT-5.2.
**Problem and protocol (paraphrased).**
- Multiple test cases.
- Query types: `type i` (read a string reply) and `multi j` (read an integer reply), with a strict per-test query budget ($\leq 10$ queries).
- Final answer per test: print `answer k` and continue to the next test.
- Flushing is required for both queries and answers.

**Full statement (source). Problem.** There are $n$ flowers arranged in a line and numbered $1, \ldots, n$ from left to right. Each flower is either a lily or a rose. For an integer $j$ with $0 \leq j \leq n$, let $l_j$ denote the number of lilies among the leftmost $j$ flowers, and let $r_j$ denote the number of roses among the rightmost $n - j$ flowers.

Initially, only $n$ is provided; the flower types are hidden. Your task is to find an integer $k$ ($0 \leq k \leq n$) such that $l_k = r_k$ using a limited number of queries. At least one such $k$ is guaranteed to exist.

**Queries.** In one query you may perform one of the following:

- `type i` ($1 \le i \le n$): the interactor returns `lily` or `rose`, the type of flower $i$.
- `multi j` ($0 \le j \le n$): the interactor returns the integer value $l_j \cdot r_j$.

**Interaction.**
- Read an integer $t$ ($1 \le t \le 100$), the number of test cases.
- For each test case, read $n$ ($1 \le n \le 100$), then issue queries.
- You may ask at most 10 queries per test case (Type and Multiply queries combined).
- When you have found a valid $k$, output `answer k` (this does not count as a query) and continue to the next test case.
- After processing all test cases, terminate.

**Notes.** The flower types are fixed before the interaction starts (non-adversarial / non-adaptive). Flush the output buffer after each query and each answer line. The original statement provides sample interactions, figures, and a local testing tool.

**Incorrect code excerpt (model-generated C++).**

```cpp
ios::sync_with_stdio(false);
cin.tie(nullptr);
...
cout << "answer " << finalCandidates[0] << "\n";  // BUG: no flush
// proceeds to read the next test case
```

**Harness evidence.** The run stalls until the idle timeout with no CPU activity:

```
queries=2
cpu_ms=0
```

**Why it fails.** Without flushing, the interactor never receives the `answer` line and therefore does not advance to the next test case. Meanwhile, the solver blocks on reading the next input, creating a classic interactive deadlock.

---

### QLE: query-budget overrun by exhaustive enumeration

**Source task.** 2025 Polish Collegiate Programming Contest, AMPPZ. **Model.** DeepSeek-V3.2.

**Problem and protocol (paraphrased).**
- Two-phase interaction: a large "training" phase followed by an "answering" phase.
- Training query: print `? ver s` and read an integer result, capped by a strict budget ($\le 300{,}000$ queries).
- Switch phase by printing `!`, then answer $q$ offline queries ($q \le 10{,}000$) by printing integers.

**Full statement (source). Problem.** The game "Plumber Mariusz" consists of $n + 1$ levels (numbered 0 to $n$), each with $k$ versions (numbered 0 to $k - 1$). Each version of every level (except the last) ends with two pipes, left and right, and Mariusz must choose one. Each pipe contains some number of coins and leads to a version of the next level. Exactly two pipes lead into each version of each level (except level 0); equivalently, each node in levels $> 0$ has in-degree 2.

A playthrough starts by choosing a start version `ver` at level 0 and then choosing a length-$n$ string `s` over `L` and `P` (left/right). The result is the total number of coins collected along the $n$ chosen pipes. The path itself is not revealed. Coins reset after each playthrough, so playthroughs are independent.

You may perform up to 300,000 test playthroughs and observe their results. After that, you will receive up to 10,000 queries about playthroughs; your task is to output their results. The test is fixed before the interaction begins (non-adaptive).

**Interaction protocol.**
- Read integers $n$ and $k$ ($1 \le n, k \le 20$).
- Training phase (up to 300,000 times): output `? ver s` where $0 \le \text{ver} < k$ and `s` is a length-$n$ string over `L/P`; flush; read one integer reply $x$ (the playthrough result).
- End the training phase by outputting a single `!` and flushing.
- Read an integer $q$ ($1 \le q \le 10{,}000$).
- For each of the $q$ queries, read `ver` and `s` (same format), output the integer result, and flush.

**Notes.** Flush after each line and do not open files. The original statement provides examples, sample solutions, and local testing artifacts.

**Incorrect code excerpt (model-generated C++).**

```cpp
void brute_force_all() {
  for (int ver = 0; ver < k; ver++) {
    for (int mask = 0; mask < (1 << n); mask++) {
      string s = ...;  // all 2^n strings over {L,P}
      cout << "? " << ver << " " << s << endl;
      int res;
      cin >> res;
      memo[{ver, s}] = res;
    }
  }
}
```

**Harness evidence.** The run exceeds the training budget:

```
queries=1048576  query_limit=300000
```

**Why it fails.** Enumerating all action strings scales as $k \cdot 2^n$ and quickly breaks the interaction budget. InteractBench surfaces this as QLE even when the per-query protocol is correct.

---

## RE: stream desynchronization from skipping judge feedback

**Source task.** 2025 ICPC Chengdu Regional Contest. **Model.** Gemini-2.5-Flash.
**Problem and protocol (paraphrased).**
- Multiple test cases.
- Query: print ?   x  y and read a numeric reply.
- Final answer: print !   d; the interactor then prints a one-word verdict (Correct or Wrong) before the next test begins.
- The per-test query budget is small ($\leq 5$).

**Full statement (source). Problem.** Panda is trapped by a regular polygon with $N$ vertices numbered $1, \ldots, N$ counterclockwise. The polygon's centroid must be destroyed. Panda does not know the polygon's size or his own position. He may perform at most 5 measurements. In a measurement, Panda chooses two vertices $x$ and $y$ and receives the area of the triangle formed by his position and these vertices. Your task is to output the distance from Panda's position to the centroid.

**Interaction protocol.**
- Read an integer $T$ ($1 \leq T \leq 10^4$), the number of test cases.
- For each test case, read an integer $N$ ($4 \leq N \leq 200$). Hidden parameters remain fixed during the test case. The polygon's circumradius is in $[1, 100]$, and the answer does not exceed $10,000$.
- For a measurement, output ?   x  y with $1 \leq x, y \leq N$ and $x \neq y$, flush, and read a non-negative real number (the triangle area) printed with at least 10 valid digits. If you make more than 5 queries, the interactor outputs $-1$ and terminates.
- After at most 5 measurements, output !   d where $d$ is a non-negative real number. Let $p$ be your output and $q$ be the judge answer. The answer is accepted if

$$\min\left(\left|\frac{p-q}{q}\right|, |p-q|\right) \leq 10^{-4}.$$

- After your answer, the interactor prints Correct and continues to the next test case if accepted; otherwise it prints Wrong and terminates.

**Notes.** Each output must end with a newline and be flushed (e.g., cout.flush() in C++, System.out.flush() in Java). The original statement provides an example interaction.

**Incorrect code excerpt (model-generated C++).**

```cpp
// prints the final answer but never reads the judge's "Correct/Wrong" line
cout << "! " << fixed << setprecision(10) << d << endl;
return;  // returns to the main loop
```

```
int T;
cin >> T;
while (T--) solve();
```

**Harness evidence.** The interactor terminates abnormally (broken pipe):

```
exit_codes[solution]=0
exit_codes[interactor]=-13
```

**Why it fails.** Skipping the `Correct`/`Wrong` line shifts the input stream. The next test case reads the wrong token, the solver exits early, and the interactor crashes when writing to a closed pipe, which is recorded as `RE`.

---

**TLE: unbounded pre-processing before issuing any query**

**Source task.** 2025 ICPC Asia Bangkok Regional Contest. **Model.** GPT-OSS-120B.
**Problem and protocol (paraphrased).**
- Hidden permutation $P$ over $N$ elements.
- Query: print a permutation ? q1 q2 ... qN, then read $N$ bits describing the post-meeting "sus" statuses.
- Total number of meetings is capped by a strict budget ($\leq 400$).

**Full statement (source). Problem.** There are $N$ crewmates. Each crewmate secretly suspects exactly one crewmate, and no crewmate is suspected by more than one person. Formally, there is a hidden permutation $P$ of length $N$ where $P[i]$ is the crewmate suspected by crewmate $i$. Your goal is to determine $P$ using at most $400$ emergency meetings. In one emergency meeting you choose the order in which crewmates speak. When a crewmate speaks, they accuse the crewmate they suspect. If the speaker has not yet been marked *sus*, their accusation is processed and the accused crewmate becomes sus. If the speaker is already sus, their accusation is ignored. After the meeting, the ship reports the sus status of all crewmates as a binary array. Each meeting is independent: all crewmates start as not sus.

**Interaction protocol.**
- Read an integer $N$ ($2 \leq N \leq 100$).
- A query has the form ? q1 q2 ... qN, where $(q_1, \ldots, q_N)$ is a permutation of $\{1, \ldots, N\}$ indicating the speaking order.
- After each query, flush and read a line containing $N$ integers $s_1, \ldots, s_N$. Here $s_i = 1$ means crewmate $i$ is sus after the meeting, and $s_i = 0$ means not sus.
- You may perform at most $400$ queries.
- When you have determined $P$, output ! p1 p2 ... pN and terminate immediately.

**Notes.** The interactor is non-adaptive: the hidden permutation does not depend on your queries. Remember to flush after each query and after the final answer; otherwise you may receive an idleness verdict. The original statement provides example interactions and additional notes.

**Incorrect code excerpt (model-generated C++).**

```
mt19937 rng(...);
while (true) {
  // generate random permutations until a rare property holds
  for (int t = 0; t < K; ++t) {
    vector<int> a(N);
    iota(a.begin(), a.end(), 1);
    shuffle(a.begin(), a.end(), rng);
    ...
  }
  if (ok) break;  // BUG: no iteration cap, no fallback
}
// TLE happens before the first interactive query.
```

**Harness evidence.** The solver times out without issuing any query:

```
queries=0
cpu_ms=19990
```

**Why it fails.** The submission spends the entire time budget in uncontrolled randomized pre-processing. In interactive tasks, computation before the first query counts toward the same time limit, so such unbounded setup loops lead to

```
TLE.
```

---

**MLE: stack growth from a missing empty-partition check**

**Source task.** 2025 ICPC Asia Pacific Championship. **Model.** DeepSeek-R1-Distill-Llama-70B.
**Problem and protocol (paraphrased).**
- The judge holds a hidden expression over $n$ occurrences of $x$ using a binary operator $(a - b)$ on bits.
- Query: print `query S` where $S$ is a binary string of length $n$, then read a reply bit.
- The query budget is 500.
- Final answer: print `answer T` and terminate.

**Full statement (source). Problem.** The judge holds a hidden expression over $n$ terminal symbols $x$ using the binary operator $(a - b)$ on bits: $(a - b) = 1$ if $a = 1$ and $b = 0$, and $(a - b) = 0$ otherwise. The expression is a binary tree whose leaves are occurrences of $x$. At the start you are given $n$; the expression is hidden.
**Queries.** In one query, you provide a binary string $S$ of length $n$. The judge substitutes the $i$-th occurrence of $x$ (from the left) with $S_i$ and evaluates the expression, returning 0 or 1.
**Interaction protocol.**
- Read $n$ ($3 \leq n \leq 200$).
- To ask a query, print `query S`, flush, and read one integer reply (0 or 1).
- You may ask at most 500 queries.
- When you have identified the expression, print `answer T` and terminate.

**Notes.** The judge is non-adaptive: the hidden expression is fixed before the interaction starts. Flush the output buffer after each output line.
**Incorrect code excerpt (model-generated C++).**

```cpp
// Partition x_list based on one-hot query replies.
vector<int> left, right;
for (int i = 0; i < (int)x_list.size(); ++i) {
  if (results[i]) left.push_back(x_list[i]);
  else right.push_back(x_list[i]);
}
// BUG: does not handle left.empty() or right.empty().
string left_expr = build_expression(left, n);
string right_expr = build_expression(right, n);
```

**Harness evidence.** The run exceeds the memory cap and crashes:

```
queries=399  query_limit=500
memory_limit_mb=256
max_rss_solution_kb=265472
exit_codes[solution]=-11
stderr_interactor_tail=wrong answer Unexpected EOF (solution terminated without
    providing answer)
```

**Why it fails.** When the partition produces an empty `left` or `right` set, the recursion has no base case and keeps expanding, causing stack growth and eventually a crash. We classify such runs as MLE when peak RSS exceeds the task's memory limit; the interactor may subsequently report WA due to premature termination (e.g., `Unexpected EOF`).

---

**WA: algorithmic error despite protocol compliance**

**Source task.** 2025 ICPC Rocky Mountain Regional Contest. **Model.** GPT-4.1.
**Problem and protocol (paraphrased).**
- There are $N$ applicants with a hidden strict total order.
- Query: print `? a b` and read the winner, capped by a linear budget ($\leq 12N + 1000$).
- Output: print `!` followed by a set of $K$ applicants that must equal the true top-$K$.

**Full statement (source). Interactive note.** This is an interactive problem: the input depends on your program's output.

**Problem.** There are $N$ applicants, each with a distinct skill level (no ties). You may compare two applicants by holding a match; the stronger one wins. Skill levels are transitive: if $a$ defeats $b$ and $b$ defeats $c$, then $a$ defeats $c$. Given $K$, your task is to determine the $K$ strongest applicants using a limited number of matches.

**Interaction.**
- Read integers $N$ and $K$ ($1 \leq N \leq 15000$, $1 \leq K \leq \min(N, 30)$).
- You may make at most $12N + 1000$ queries.
- A query has the form ? a b with $1 \leq a, b \leq N$ and $a \neq b$. Flush and read the reply, either $a$ or $b$, indicating the winner.
- Once you have identified the top-$K$ applicants, output ! x1 x2 ... xK (any order) and terminate.

**Non-adaptivity.** The interactor is not adaptive: the skill levels are fixed at the start.

**Uniqueness requirement.** In addition to being correct, your queries must establish that your output is the unique possible set of top-$K$ applicants consistent with all observed comparisons; otherwise the verdict is Wrong Answer.

**Invalid input.** If the interactor receives invalid input, it outputs $-1$ and terminates; your program should exit cleanly.

**Local testing.** The original statement provides a command-line tool for local testing.

**Incorrect code excerpt (model-generated C++).**

```cpp
// If K > selected.size(), fill up with random (this should never happen).
while ((int)selected.size() < K) {
  for (int i = 1; i <= N && (int)selected.size() < K; ++i) {
    selected.insert(i);
  }
}
cout << "! ";
for (int x : selected) cout << x << " ";
cout << endl;
```

**Harness evidence.** The interactor rejects the final set:

```
wrong answer Incorrect set of applicants (not the actual top K)
```

**Why it fails.** The submission is protocol-correct and stays within the query budget, but its selection logic does not guarantee that the reported set equals the true top-$K$. This is a reasoning-level failure: interactive problems often require sufficient comparison evidence to uniquely certify the answer, not merely to "guess" a plausible set.

### D.3. End-to-End Task Showcase: `cf2036G`

To make the offline evaluation pipeline concrete, we provide one end-to-end interactive task example using `cf2036G` (Codeforces; non-adaptive). This showcase illustrates how a task is packaged (protocol, generator, local interactor, and offline hidden cases) and evaluated with recorded artifacts.

**Protocol excerpt (interaction-relevant clauses).** We summarize the interaction rules from the original statement and omit narrative content:

- **Hidden multiset.** Exactly three distinct values $a, b, c$ appear once; every other value in $[1, n]$ appears twice. The goal is to output $(a, b, c)$ in any order.
- **Query.** Output xor l r with $1 \leq l \leq r \leq n$, flush, and read one integer reply (the XOR of all values in $[l, r]$).
- **Budget.** At most 150 queries per test case; further queries return $-1$; the solver should terminate upon receiving $-1$.
- **Answer.** Output ans a b c (any order), then proceed to the next test case.
- **Non-adaptive.** The hidden input is fixed in advance and does not depend on the solver's queries.

**Test-case generator (full listing).**

**cf2036G generator (`gen_cases.cpp`)**

```cpp
#include "testlib.h"
#include <bits/stdc++.h>
using namespace std;
```

```cpp
static long long highestPow2LE(long long n) {
    long long p = 1;
    while ((p << 1) > 0 && (p << 1) <= n) p <<= 1;
    return p;
}

static long long pickNotIn(long long n, const vector<long long>& used) {
    while (true) {
        long long x = rnd.next(1LL, n);
        bool ok = true;
        for (long long u : used) if (x == u) { ok = false; break; }
        if (ok) return x;
    }
}

static void ensureDistinctInRange(long long n, long long &a, long long &b, long long &
    c) {
    auto inRange = [&](long long x) { return 1LL <= x && x <= n; };
    if (!inRange(a)) a = rnd.next(1LL, n);
    if (!inRange(b)) b = rnd.next(1LL, n);
    if (!inRange(c)) c = rnd.next(1LL, n);

    if (a == b) b = pickNotIn(n, {a});
    if (a == c) c = pickNotIn(n, {a, b});
    if (b == c) c = pickNotIn(n, {a, b});
}

static void genTriple(long long n, long long seed, long long &a, long long &b, long
    long &c) {
    // Hand-crafted corner cases for small seeds.
    if (seed == 1) {
        a = 1; b = 2; c = 3; // minimal n=3, and a^b^c = 0
        return;
    }
    if (seed == 2) {
        a = 1;
        b = n - 1;
        c = n; // boundary-heavy, (n-1)^n = 1 => total xor = 0
        return;
    }
    if (seed == 3) {
        long long x = (1LL << 58);
        a = 1;
        b = x;
        c = x + 1; // x^(x+1)=1 => total xor = 0, mixes low/high bits
        if (c > n) {
            // Fallback: still keep xor=0 if possible.
            a = n;
            b = n - 1;
            c = 1;
        }
        return;
    }

    // Random stress + adversarial structured patterns for other seeds.
    int type = rnd.next(0, 99);

    if (type < 50) {
        // Adversarial: near power-of-two boundary with total xor = 0.
        long long k = highestPow2LE(n);
        a = k;
        b = k - 1;
        c = 1;
```

```cpp
        if (k <= 2) {
            // For very small n, just pick distinct.
            a = 1; b = 2; c = 3;
        } else if (b == c) {
            c = 2;
        }
        ensureDistinctInRange(n, a, b, c);
        return;
    }

    if (type < 75) {
        // Adversarial: force a^b^c = 0 if feasible.
        bool ok = false;
        for (int it = 0; it < 200; it++) {
            long long x = rnd.next(1LL, n);
            long long y = rnd.next(1LL, n);
            if (x == y) continue;
            long long z = x ^ y;
            if (1LL <= z && z <= n && z != x && z != y) {
                a = x; b = y; c = z;
                ok = true;
                break;
            }
        }
        if (ok) return;
        // Fallback to spread.
    }

    // Spread across the range to maximize branching/partition uncertainty.
    long long third1 = max(1LL, n / 3);
    long long third2 = max(third1 + 1, (2 * n) / 3);

    a = rnd.next(1LL, third1);
    b = rnd.next(min(third1 + 1, n), min(third2, n));
    c = rnd.next(min(third2 + 1, n), n);

    ensureDistinctInRange(n, a, b, c);
}

int main(int argc, char** argv) {
    registerGen(argc, argv, 1);

    string mode = opt<string>("mode", "non");
    long long fixedN = opt<long long>("n", 0LL);
    long long nmin = opt<long long>("nmin", 3LL);
    long long nmax = opt<long long>("nmax", 1000000000000000000LL);

    ensuref(nmin >= 3, "nmin must be >= 3");
    ensuref(nmax >= nmin, "nmax must be >= nmin");

    long long seed = 1;
    if (argc > 1) seed = atoll(argv[1]);

    long long n;
    if (fixedN != 0) {
        n = fixedN;
    } else if (seed == 1) {
        n = 3;
    } else if (seed == 2) {
        n = 1000000000000000000LL;
    } else if (seed == 3) {
        n = 999999999999999937LL;
    } else {
```

```
        long long span = nmax - nmin;
        long long delta = 0;
        if (span > 0) delta = rnd.next(0LL, min(1000000LL, span));
        n = nmax - delta;
        if (n < nmin) n = nmin;
        if (n > nmax) n = nmax;
    }

    ensuref(3LL <= n && n <= 1000000000000000000LL, "n out of bounds");

    if (mode == "non") {
        long long a = 1, b = 2, c = 3;
        genTriple(n, seed, a, b, c);
        ensureDistinctInRange(n, a, b, c);
        ensuref(a != b && a != c && b != c, "a,b,c must be distinct");
        cout << 1 << "\n";
        cout << n << "\n";
        cout << a << " " << b << " " << c << "\n";
        return 0;
    }

    if (mode == "adp") {
        // Adaptive schema: only provide n; the interactor picks a,b,c adversarially
    within [1..n].
        cout << 1 << "\n";
        cout << n << "\n";
        return 0;
    }

    quitf(_fail, "Unknown --mode (expected non/adp): %s", mode.c_str());
}
```

**Local interactor (full listing).**

**cf2036G non-adaptive interactor (`non_adaptive.cpp`)**

```cpp
#include "testlib.h"
#include <bits/stdc++.h>
using namespace std;

static long long queries = 0;
static long long query_limit = 0;

static void log_metrics() {
    try {
        tout << "queries=" << queries << "\n";
        tout << "query_limit=" << query_limit << "\n";
        tout.flush();
    } catch (...) {
        // best-effort
    }
}

static void finish(TResult verdict, const string& msg) {
    log_metrics();
    quitf(verdict, "%s", msg.c_str());
}

static bool readLongLongFromCin(long long &out) {
    // Robust-ish: rely on operator>>; if it fails, caller handles.
    cin >> out;
```

```cpp
    return (bool)cin;
}

int main(int argc, char** argv) {
    registerInteraction(argc, argv);
    atexit(log_metrics);

    long long t = 0;
    try {
        t = inf.readLong();
    } catch (...) {
        // In practice testlib terminates the process on read errors, but keep a
    fallback.
        query_limit = 0;
        log_metrics();
        return 0;
    }
    if (t < 1 || t > 300) {
        query_limit = 0;
        log_metrics();
        quitf(_fail, "Invalid t in hidden input: %lld", t);
    }

    struct CaseData {
        long long n, a, b, c;
    };
    vector<CaseData> cases;
    cases.reserve((size_t)t);

    for (long long i = 0; i < t; i++) {
        long long n = inf.readLong();
        long long a = inf.readLong();
        long long b = inf.readLong();
        long long c = inf.readLong();
        cases.push_back({n, a, b, c});
    }

    query_limit = 150LL * t;
    log_metrics(); // MUST happen before any blocking reads from contestant.

    const long long hard_cap = max(200000LL, 10LL * query_limit);

    cout << t << "\n";
    cout.flush();

    for (long long tc = 0; tc < t; tc++) {
        const long long n = cases[tc].n;
        const long long A = cases[tc].a;
        const long long B = cases[tc].b;
        const long long C = cases[tc].c;

        if (!(3LL <= n && n <= 1000000000000000000LL)) finish(_fail, "Invalid n in
    hidden input");
        auto inRange = [&](long long x) { return 1LL <= x && x <= n; };
        if (!inRange(A) || !inRange(B) || !inRange(C) || A == B || A == C || B == C)
            finish(_fail, "Invalid (a,b,c) in hidden input");

        cout << n << "\n";
        cout.flush();

        long long case_queries = 0;
        while (true) {
            if (queries > hard_cap) finish(_pe, "Hard cap exceeded");
```

```
            string cmd;
            if (!(cin >> cmd)) finish(_pe, "Unexpected EOF from contestant");

            if (cmd == "xor") {
                long long l, r;
                if (!readLongLongFromCin(l) || !readLongLongFromCin(r))
                    finish(_pe, "Malformed xor query (expected two integers)");

                if (!(1LL <= l && l <= r && r <= n))
                    finish(_pe, "Invalid xor query bounds");

                queries++;
                case_queries++;

                if (case_queries > 150) {
                    cout << -1 << "\n";
                    cout.flush();
                    continue;
                }

                long long res = 0;
                if (l <= A && A <= r) res ^= A;
                if (l <= B && B <= r) res ^= B;
                if (l <= C && C <= r) res ^= C;

                cout << res << "\n";
                cout.flush();
            } else if (cmd == "ans") {
                long long x, y, z;
                if (!readLongLongFromCin(x) || !readLongLongFromCin(y) || !
    readLongLongFromCin(z))
                    finish(_pe, "Malformed ans (expected three integers)");

                if (!inRange(x) || !inRange(y) || !inRange(z) || x == y || x == z || y
     == z)
                    finish(_wa, "Invalid answer values");

                array<long long, 3> got = {x, y, z};
                array<long long, 3> want = {A, B, C};
                sort(got.begin(), got.end());
                sort(want.begin(), want.end());

                if (got != want) finish(_wa, "Wrong answer");

                if (tc + 1 == t) finish(_ok, "OK");
                break; // proceed to next test
            } else {
                finish(_pe, "Unknown command (expected 'xor' or 'ans')");
            }
        }
    }

    finish(_pe, "Contestant did not finish all testcases");
    return 0;
}
```

**Hidden offline test case (example).**  The offline suite provides hidden inputs consumed by the local interactor (not by the solver).

**Hidden case example (`cases/003.in`)**

```
1
99999999999999937
1 288230376151711744 288230376151711745
```

**Representative passing solver (model-generated).**

**Passing solver (C++; GPT-5.2)**

```cpp
#include <bits/stdc++.h>
using namespace std;

using ull = unsigned long long;
using ll = long long;

static ull N;

static ull ask_xor(ull l, ull r) {
    cout << "xor " << l << " " << r << endl; // endl flushes
    ll resp;
    if (!(cin >> resp)) exit(0);
    if (resp == -1) exit(0);
    return (ull)resp;
}

static ull prefix_xor(ull x) {
    if (x == 0) return 0;
    return ask_xor(1, x);
}

int main() {
    ios::sync_with_stdio(false);
    cin.tie(nullptr);

    int t;
    cin >> t;
    while (t--) {
        ull n;
        cin >> n;
        N = n;

        if (n == 3) {
            cout << "ans 1 2 3" << endl;
            continue;
        }

        ull total = ask_xor(1, n);

        if (total != 0) {
            // Find smallest stolen number s1: first x where prefix_xor(x) != 0.
            ull lo = 0, hi = n;
            while (lo + 1 < hi) {
                ull mid = lo + (hi - lo) / 2;
                ull v = prefix_xor(mid);
                if (v == 0) lo = mid;
                else hi = mid;
            }
            ull s1 = hi;

            // Find largest stolen number s3: first x where prefix_xor(x) == total.
```

```
            lo = s1;
            hi = n;
            while (lo + 1 < hi) {
                ull mid = lo + (hi - lo) / 2;
                ull v = prefix_xor(mid);
                if (v == total) hi = mid;
                else lo = mid;
            }
            ull s3 = hi;

            // Get prefix before s3 to recover s2.
            ull pref_before = prefix_xor(s3 - 1); // = s1 ^ s2
            ull s2 = s1 ^ pref_before;

            cout << "ans " << s1 << " " << s2 << " " << s3 << endl;
        } else {
            // total == 0: find a stolen number by scanning MSB blocks [2^k, 2^{k
    +1}-1].
            // In this case, not all three can share the same MSB, so some block has
    exactly one.
            ull known = 0;
            for (int k = 0; k <= 60; k++) {
                ull l = 1ULL << k;
                if (l > n) break;
                ull r = min(n, (1ULL << (k + 1)) - 1ULL);
                ull v = ask_xor(l, r);
                if (v != 0 && (v & (1ULL << k)) != 0) {
                    known = v; // singleton in this MSB block -> v is that number
                    break;
                }
            }
            if (known == 0) exit(0); // should be impossible; safety

            // Now the remaining two numbers y,z satisfy y ^ z = known (since total=0)
    .
            // Remove 'known' from any prefix query by XORing it back if it's included
    .
            auto pref2 = [&](ull x) -> ull {
                ull v = prefix_xor(x);
                if (known <= x) v ^= known;
                return v; // xor of remaining two within [1..x]
            };

            // Find the smaller of the remaining two: first x where pref2(x) != 0.
            ull lo = 0, hi = n;
            while (lo + 1 < hi) {
                ull mid = lo + (hi - lo) / 2;
                ull v = pref2(mid);
                if (v == 0) lo = mid;
                else hi = mid;
            }
            ull y = hi;
            ull z = known ^ y;

            cout << "ans " << known << " " << y << " " << z << endl;
        }
    }
    return 0;
}
```

**Failure counterpart (model-generated).**

**Failure solver (C++; GPT-4.1)**

```cpp
#include <iostream>
#include <vector>
#include <algorithm>
using namespace std;

// Helper: XOR from 1 to x
long long xor1toN(long long n) {
    long long res;
    if (n % 4 == 0) res = n;
    else if (n % 4 == 1) res = 1;
    else if (n % 4 == 2) res = n + 1;
    else res = 0;
    return res;
}

int main() {
    ios::sync_with_stdio(false);
    cin.tie(nullptr);

    int t;
    cin >> t;
    while (t--) {
        long long n;
        cin >> n;

        // Query: "xor 1 n" to get the overall xor
        cout << "xor 1 " << n << endl;
        cout.flush();

        long long S;
        cin >> S;
        if (S == -1) return 0;

        // Compute double xor of 1..n
        long long doubleXor = xor1toN(n) ^ xor1toN(n); // always 0

        // The result S = doubleXor ^ (a ^ b ^ c) = 0 ^ (a ^ b ^ c)
        // So S = a ^ b ^ c

        long long abcXor = S;

        // Now, we need to find a, b, c such that a ^ b ^ c = abcXor, and a, b, c are
    distinct in [1,n]
        // Let's try to get more info.
        // Query: "xor 1 1", "xor 2 2", "xor 3 3"
        // To find which ones have only 1 copy (stolen ones will return their number,
    others will return 0)

        vector<long long> candidates;
        int cnt_queries = 1;

        for (long long i = 1; i <= min(50LL, n); ++i) {
            cout << "xor " << i << " " << i << endl;
            cout.flush();
            cnt_queries++;
            long long ans;
            cin >> ans;
            if (ans == -1) return 0;
            if (ans != 0) {
                candidates.push_back(i);
                if (candidates.size() == 3) break;
```

```
        }
    }

    if (candidates.size() == 3) {
        cout << "ans " << candidates[0] << " " << candidates[1] << " " <<
candidates[2] << endl;
        cout.flush();
        continue;
    }

    // If n is very large, it's likely that the missing ones are not in the first
50.
    // Let's query last 50 as well
    for (long long i = max(1LL, n - 49); i <= n; ++i) {
        if (cnt_queries >= 150) break;
        cout << "xor " << i << " " << i << endl;
        cout.flush();
        cnt_queries++;
        long long ans;
        cin >> ans;
        if (ans == -1) return 0;
        if (ans != 0) {
            // Don't add duplicates
            if (find(candidates.begin(), candidates.end(), i) == candidates.end())
 {
                candidates.push_back(i);
                if (candidates.size() == 3) break;
            }
        }
    }

    if (candidates.size() == 3) {
        cout << "ans " << candidates[0] << " " << candidates[1] << " " <<
candidates[2] << endl;
        cout.flush();
        continue;
    }

    // If still not found, need to search more!
    // Try blocks: step by 1000000, query "xor i i"
    long long step = max(n / 100, 1LL);
    for (long long i = 1; i <= n && cnt_queries < 150; i += step) {
        cout << "xor " << i << " " << i << endl;
        cout.flush();
        cnt_queries++;
        long long ans;
        cin >> ans;
        if (ans == -1) return 0;
        if (ans != 0) {
            if (find(candidates.begin(), candidates.end(), i) == candidates.end())
 {
                candidates.push_back(i);
                if (candidates.size() == 3) break;
            }
        }
    }

    // If not found, try blocks in the middle
    if (candidates.size() < 3 && n > 100) {
        long long mid = n / 2;
        for (long long off = -25; off <= 25 && cnt_queries < 150; ++off) {
            long long idx = mid + off;
            if (idx < 1 || idx > n) continue;
```

```
                cout << "xor " << idx << " " << idx << endl;
                cout.flush();
                cnt_queries++;
                long long ans;
                cin >> ans;
                if (ans == -1) return 0;
                if (ans != 0) {
                    if (find(candidates.begin(), candidates.end(), idx) == candidates.
    end()) {
                        candidates.push_back(idx);
                        if (candidates.size() == 3) break;
                    }
                }
            }
        }

        // If still not found, as a last resort, brute force up to 150 queries
        for (long long i = 1; i <= n && cnt_queries < 150; ++i) {
            cout << "xor " << i << " " << i << endl;
            cout.flush();
            cnt_queries++;
            long long ans;
            cin >> ans;
            if (ans == -1) return 0;
            if (ans != 0) {
                if (find(candidates.begin(), candidates.end(), i) == candidates.end())
    {
                    candidates.push_back(i);
                    if (candidates.size() == 3) break;
                }
            }
        }

        // Now, output the answer
        if (candidates.size() == 3) {
            cout << "ans " << candidates[0] << " " << candidates[1] << " " <<
    candidates[2] << endl;
            cout.flush();
        } else {
            // If unable to find, output 1 2 3 (possible for very small n)
            cout << "ans 1 2 3" << endl;
            cout.flush();
        }
    }

    return 0;
}
```

**Failure reason (statement-aligned).** The failure solver relies heavily on point queries of the form `xor i i` and sparse scanning heuristics to locate $a, b, c$. This probing pattern can fail to locate values that lie far from the probed indices when $n$ is large. On the bundled hidden case (`cases/003.in`), two of the three target values are very large (around $2^{58}$) and lie far from the probed indices, so they are not discovered within the query budget, leading to an incorrect reconstructed triple and a WA verdict.

**Evaluation summary.** We evaluate offline on the bundled 100-case suite (files under `cases/`) using the same two-process local runner as in the main experiments. In this bundle, the passing solver succeeds on all cases, while the failure solver produces a WA on `cases/003.in` after exhausting the per-case query budget. For brevity, we omit raw logs and machine-readable result dumps here.

## D.4. Adaptive Interactor Example: `cf2096G`

To complement the non-adaptive end-to-end showcase in Appendix D.3, we provide an adaptive interactor example using `cf2096G` (Codeforces). The goal is not to repeat the full packaging pipeline, but to concretize what we mean by *history-dependent choices by the interactor* and how such interactors are implemented in our offline harness.

**Task and protocol (paraphrased).** The task hides an integer $x \in [1, n]$. The solver must first output the number of queries $q$, then output exactly $q$ queries. Each query is a line `k a1 a2 ...ak` where $k$ is even and the $a_i$ are distinct in $[1, n]$: if $x$ lies in the first half, the reply is `L`; if it lies in the second half, the reply is `R`; otherwise, the reply is `N`. After all $q$ queries are printed, the solver reads a length-$q$ reply string over $\{\texttt{L,R,N,?}\}$, where exactly one position is `?` (an ignored query). The solver then outputs its final guess $x$. Importantly, the statement requires using exactly the minimal number of queries $f(n)$; using more queries is rejected even if the final guess is correct.

**What "adaptive" means in this task.** Unlike a non-adaptive interactor that fixes $x$ before interaction, `cf2096G` explicitly allows the interactor to *choose* a consistent hidden value and which query to ignore *after seeing the full set of queries*, as long as at least one $x$ remains consistent with the returned reply string. Therefore, a correct solver must ensure that its query design uniquely determines $x$ even under statement-permitted history-dependent choices by the interactor (e.g., an adversarial choice of the ignored position).

**Hidden offline input (example).** In our offline harness, the hidden input for this task contains only public parameters (e.g., $n$) and does not fix a concrete hidden value in advance.

---

**Hidden case example (`cases/001.in`)**

```
1
2
```

---

**Local adaptive interactor (full listing).** Our local adaptive interactor instantiates the statement-permitted adaptivity via a deterministic worst-case policy: it selects the ignored query index that maximizes the remaining ambiguity among candidates and uses a fixed tie-breaking rule to keep evaluation reproducible.

**cf2096G adaptive interactor (`interactor_adaptive.cpp`)**

```cpp
#include "testlib.h"
#include <bits/stdc++.h>
using namespace std;

static long long queries = 0;
static long long query_limit = 0;

static void log_metrics() {
    try {
        tout << "queries=" << queries << endl;
        tout << "query_limit=" << query_limit << endl;
    } catch (...) {
        // best-effort logging only
    }
}

static void finish(TResult verdict, const string& msg) {
    log_metrics();
    quitf(verdict, "%s", msg.c_str());
}

static bool readLongLong(long long& out) {
    if (!(cin >> out)) return false;
```

```
    return true;
}

static int f_min_queries(int n) {
    // Smallest q such that 3^(q-1) >= n
    long long pw = 1;
    int q = 1;
    while (pw < n) {
        pw *= 3;
        q++;
        if (q > 60) break;
    }
    return q;
}

static int symToDigit(char c) {
    if (c == 'L') return 0;
    if (c == 'R') return 1;
    return 2; // 'N'
}

int main(int argc, char** argv) {
    registerInteraction(argc, argv);
    atexit(log_metrics);

    int t = 0;
    try {
        t = inf.readInt();
    } catch (...) {
        finish(_pe, "failed to read t from input");
    }
    if (t < 1 || t > 100000) finish(_pe, "t out of reasonable range");

    query_limit = 20LL * t;
    log_metrics(); // must happen before any blocking read from std::cin

    cout << t << "\n";
    cout.flush();

    const long long hard_cap = max(200000LL, 10LL * query_limit);

    for (int tc = 1; tc <= t; tc++) {
        int n = 0;
        try {
            n = inf.readInt();
        } catch (...) {
            finish(_pe, "failed to read n from input");
        }
        if (n < 2 || n > 200000) finish(_pe, "n out of range");

        cout << n << "\n";
        cout.flush();

        long long qll = 0;
        if (!readLongLong(qll)) finish(_pe, "failed to read q from contestant");
        if (qll < 1 || qll > 20) finish(_pe, "q must be in [1,20]");
        int q = (int)qll;

        int fq = f_min_queries(n);
        if (q != fq) finish(_wa, "wrong number of queries (must be minimal f(n))");

        queries += q;
        if (queries > hard_cap) finish(_pe, "too many queries (hard cap exceeded)");
```

```
        vector<vector<char>> code(n + 1, vector<char>(q, 'N'));

        for (int qi = 0; qi < q; qi++) {
            long long kll = 0;
            if (!readLongLong(kll)) finish(_pe, "failed to read k");
            if (kll < 2 || kll > n) finish(_pe, "k out of range");
            int k = (int)kll;
            if (k % 2 != 0) finish(_pe, "k must be even");

            vector<int> a(k);
            vector<int> seen(n + 1, 0);
            for (int i = 0; i < k; i++) {
                long long xll = 0;
                if (!readLongLong(xll)) finish(_pe, "failed to read array element");
                if (xll < 1 || xll > n) finish(_pe, "array element out of range");
                int x = (int)xll;
                if (seen[x]) finish(_pe, "array elements must be distinct");
                seen[x] = 1;
                a[i] = x;
            }

            int h = k / 2;
            for (int i = 0; i < h; i++) code[a[i]][qi] = 'L';
            for (int i = h; i < k; i++) code[a[i]][qi] = 'R';
        }

        int bestJ = 0;
        int bestRep = 1;
        int bestSize = -1;

        vector<pair<long long, int>> keys;
        keys.reserve(n);

        for (int j = 0; j < q; j++) {
            keys.clear();
            for (int x = 1; x <= n; x++) {
                long long key = 0;
                for (int i = 0; i < q; i++) {
                    if (i == j) continue;
                    key = key * 3 + symToDigit(code[x][i]);
                }
                keys.emplace_back(key, x);
            }
            sort(keys.begin(), keys.end(), [](const auto& p1, const auto& p2) {
                if (p1.first != p2.first) return p1.first < p2.first;
                return p1.second < p2.second;
            });

            for (int idx = 0; idx < n; ) {
                int nxt = idx + 1;
                while (nxt < n && keys[nxt].first == keys[idx].first) nxt++;
                int sz = nxt - idx;
                int rep = keys[idx].second; // smallest x in this group due to sort
                if (sz > bestSize ||
                    (sz == bestSize && (j < bestJ || (j == bestJ && rep < bestRep))))
{
                    bestSize = sz;
                    bestJ = j;
                    bestRep = rep;
                }
                idx = nxt;
            }
```

```
        }

        string s(q, 'N');
        for (int i = 0; i < q; i++) s[i] = (i == bestJ ? '?' : code[bestRep][i]);

        cout << s << "\n";
        cout.flush();

        long long xll = 0;
        if (!readLongLong(xll)) finish(_pe, "failed to read final answer x");
        if (xll < 1 || xll > n) finish(_pe, "final answer x out of range");
        int x = (int)xll;

        bool inGroup = true;
        for (int i = 0; i < q; i++) {
            if (i == bestJ) continue;
            if (code[x][i] != code[bestRep][i]) {
                inGroup = false;
                break;
            }
        }

        if (!(bestSize == 1 && inGroup)) {
            finish(_wa, "answer not uniquely determined by the interaction");
        }
    }

    finish(_ok, "OK");
    return 0;
}
```

### D.5. Mutation-based Stress Tests

**Motivation.** Section 3 introduces mutation-based stress tests—lightly mutated submissions used as an internal robustness check. Here we detail the construction procedure and mutation operators used in our construction loop.

**Two stress-test sets.** We generate 15 mutated submissions per task and partition them into two sets.

**Wrong-answer, protocol-valid submissions.** These mutated submissions follow the protocol and respect the budget, but violate task semantics so a statement-consistent evaluator should reject them. They test correctness checking under realistic interaction traces without relying on protocol violations.

**Protocol- or budget-invalid submissions.** These mutated submissions intentionally violate protocol rules or exceed query budgets to exercise edge cases of parsing, accounting, and termination. They test whether the evaluator enforces the published interaction rules consistently.

**Submission sources.** We apply lightweight source-level mutations to existing solutions. We prioritize verified accepted submissions as stable bases and optionally derive mutated submissions from representative rejected submissions to diversify failure modes.

**Mutation operators.** We use a small and interpretable set of operators.

**Wrong-answer, protocol-valid.**

- State-update perturbations that preserve I/O format but break a key invariant, such as reversing an interval update direction or skipping a required update.
- Early-stop behaviors that terminate interaction and output a guess while still following the protocol.
- Response-interpretation perturbations that keep queries valid but mis-handle interactor feedback, such as flipping a binary response or shifting a comparison threshold.

- Query-strategy coarsening that keeps the protocol valid but drops a necessary class of queries, making the strategy incomplete.
- Output perturbations for tasks with uniquely checkable answers or explicit validity constraints, where a small edit to the final output violates a requirement while keeping the protocol well-formed.

**Protocol- or budget-invalid.**

- Parameter boundary violations designed to test fencepost cases in the parser and checker.
- Malformed tokens or unsupported commands that test strict parsing.
- Budget fencepost violations that exceed the query limit by one step.
- Invalid termination behaviors, such as continuing to send output after the interactor has terminated.

**Filtering and usage.**   Mutated submissions are used as rejected stress-test submissions during validation rather than as additional officially labeled data. We require each stress-test submission to be rejected; any stress-test submission that is mistakenly accepted triggers refinement of the generator, interactor, or checkers. After refinement, we re-validate on the officially labeled human subset of the verification pool to ensure the fix does not introduce regressions.

### D.6. Expert Intervention Case Study

We include an expert-intervention case study to illustrate how a statement-level semantic mismatch in an interactor can lead to incorrect Accepted/Rejected decisions, and how expert review restores statement-consistent behavior. In our construction, 31 tasks required such expert intervention after reaching the iteration cap (Section 3). We use cf1847E to illustrate how a statement ambiguity can lead to a mismatched verdict in an initial interactor implementation, and how expert review resolves the discrepancy.

---

**cf1847E: Made in Heaven (problem statement)**

This is an interactive problem.

Made in Heaven is a rather curious Stand. Of course, it is (arguably) the strongest Stand in existence, but it is also an ardent puzzle enjoyer. For example, it gave Qtaro the following problem recently.

Made in Heaven has $n$ hidden integers $a_1, a_2, \ldots, a_n$ ($3 \leq n \leq 5000$, $1 \leq a_i \leq 4$). Qtaro must determine all the $a_i$ by asking Made in Heaven some queries of the following form.

- In one query, Qtaro gives three distinct indices $i$, $j$, and $k$.
- If $a_i$, $a_j$, $a_k$ form the sides of a non-degenerate triangle, Made in Heaven responds with the area of this triangle.
- Otherwise, Made in Heaven responds with $0$.

By asking at most 5500 such questions, Qtaro must either tell Made in Heaven all the values of the $a_i$, or report that it is not possible to uniquely determine them.

Unfortunately due to the universe reboot, Qtaro is not as smart as Jotaro. Please help Qtaro solve Made in Heaven's problem.

**Non-degenerate triangle.** Three positive integers $a, b, c$ are said to form the sides of a non-degenerate triangle if and only if all of the following three inequalities hold.

- $a + b > c$,
- $b + c > a$,
- $c + a > b$.

**Interaction.** The interaction begins with reading $n$ ($3 \leq n \leq 5000$), the number of hidden integers. To ask a question corresponding to a triple $(i, j, k)$ with distinct indices, output ?   i  j  k and then read a single integer $s$.

- If $s = 0$, then $a_i$, $a_j$, and $a_k$ are not the sides of a non-degenerate triangle.
- Otherwise, $s = 16\Delta^2$, where $\Delta$ is the area of the triangle. The area is provided in this format for convenience so that only integer input is needed.

If the numbers $a_i$ cannot be uniquely determined, output !   -1. On the other hand, if you have determined all the values of $a_i$, output !   $a_1\ a_2\ \ldots\ a_n$ on a single line.

The interactor is non-adaptive. The hidden array $a_1, a_2, \ldots, a_n$ is fixed beforehand and is not changed during the interaction process.

**Flushing.** After printing a query, do not forget to output the end of line and flush the output. Otherwise, you may get "Idleness limit exceeded". To do this, use the following methods in different languages.

- `fflush(stdout)` or `cout.flush()` in C++;
- `System.out.flush()` in Java;
- `flush(output)` in Pascal;
- `stdout.flush()` in Python;
- see the documentation for other languages.

**Protocol.** We focus on the statement clauses that matter for the `! -1` ambiguity verdict:

- **Hidden array.** The interactor fixes hidden integers $a_1, \ldots, a_n$ with $3 \leq n \leq 5000$ and $a_i \in \{1, 2, 3, 4\}$.
- **Query.** The solver prints `? i j k` with three distinct indices, flushes, and reads an integer reply $s$. The reply is 0 when the three values do not form a non-degenerate triangle, and otherwise $s = 16\Delta^2$, where $\Delta$ is the triangle area.
- **Budget.** The solver may ask at most 5500 queries.
- **Final output.** The solver prints either `! $a_1$ $a_2$ ... $a_n$` or `! -1` when it is not possible to uniquely determine the hidden array.
- **Fixed hidden array during interaction.** The hidden array remains unchanged throughout the dialogue.

**Ambiguity witness.** The solver queries triples of indices and receives an oracle reply that depends only on the corresponding triple of hidden values; the hidden array is fixed before the interaction and does not change during the dialogue. The statement allows `! -1` whenever the hidden array is *not uniquely identifiable* under the protocol: it must be Accepted if there exists another hidden array that would produce identical oracle replies for all possible queries.

Let $\text{reply}(x, y, z)$ denote the interactor response to a triple of values $(x, y, z)$. We call a hidden array $\mathbf{a}$ ambiguous if there exists another $\mathbf{b} \neq \mathbf{a}$ such that

$$\forall\, i, j, k \text{ distinct}, \quad \text{reply}(a_i, a_j, a_k) = \text{reply}(b_i, b_j, b_k).$$

In this case, output `! -1` is the only statement-consistent behavior.

**Minimal counterexample.** For $n = 4$, consider the two hidden arrays $\langle 1, 1, 1, 2 \rangle$ and $\langle 1, 1, 1, 4 \rangle$. For any triple of indices, the oracle response depends on whether the three values form a non-degenerate triangle. In both arrays, the triple consisting of the three 1s yields the same non-zero response, while any triple containing the fourth element yields 0. Therefore, the replies to all allowed queries are identical, yet the hidden arrays differ. Consequently, the hidden array is not uniquely identifiable, and `! -1` must be Accepted under the statement. This ambiguity is not an artifact of a limited query budget: when $n = 4$ there are only four possible triples, so the solver can query all of them and still cannot distinguish the two arrays.

**Non-zero replies do not imply uniqueness.** Ambiguity may persist even when the transcript contains non-zero replies. For example, $\langle 1, 2, 3, 4 \rangle$ and $\langle 1, 2, 4, 3 \rangle$ induce the same replies for all four triples when $n = 4$: all triples containing 1 return 0, and the remaining triple returns a fixed non-zero value. This shows why uniqueness cannot be inferred from the existence of any non-degenerate triangle.

**Why this affects correct solvers.** This is not a contrived edge case of a trivial program. Standard solutions explicitly output `! -1` whenever the interaction transcript admits multiple consistent hidden arrays. For example, a typical implementation enumerates candidates for small $n$ and emits `! -1` when the solution is not unique:

> **Solver excerpt: emitting `! -1` under ambiguity**
>
> ```cpp
> dfs(1);  // enumerate hidden arrays consistent with replies
> if (cnt != 1) {
>   cout << "! -1" << endl;
>   return 0;
> }
> ```

**Pre-fix and post-fix.** The pre-fix interactor incorrectly rejected `! -1` on a class of ambiguous arrays due to an overly strong uniqueness assumption. Concretely, it treated the existence of any non-degenerate triangle among the hidden values as sufficient evidence of uniqueness once $n$ is large enough, which is not guaranteed by the statement. The expert revision

replaced this heuristic with an ambiguity check aligned with the statement, leveraging the small value domain to test whether an alternative hidden array can preserve all oracle replies. After the fix, `! -1` is Accepted on ambiguous arrays while remaining Rejected on uniquely identifiable arrays, demonstrating that the correction restores semantic fidelity rather than weakening the evaluator.

*Table 13.* Effect of expert intervention on `! -1` verdicts for `cf1847E`.

| Array type with solver output `! -1` | Pre-fix | Post-fix |
|---|---|---|
| Ambiguous array (multiple consistent hidden arrays) | Rejected | Accepted |
| Uniquely identifiable array (single consistent hidden array) | Rejected | Rejected |

**Validation.** To avoid overfitting the fix, we validated the ambiguity checker by exhaustive enumeration for $n \leq 9$ over the full value domain $\{1, 2, 3, 4\}^n$. Across all enumerated arrays, every ambiguous array admitted a local witness detectable by the post-fix check (single-position replacement or two-position swap), and we observed no counterexample.

*Table 14.* Exhaustive enumeration validation of the ambiguity checker ($n \leq 9$).

| $n$ | $4^n$ | # ambiguous | # missing witness |
|---|---|---|---|
| 3 | 64 | 60 | 0 |
| 4 | 256 | 84 | 0 |
| 5 | 1024 | 135 | 0 |
| 6 | 4096 | 228 | 0 |
| 7 | 16384 | 357 | 0 |
| 8 | 65536 | 528 | 0 |
| 9 | 262144 | 747 | 0 |

---

**Interactor excerpt: pre-fix ambiguity check (incorrect)**

```
bool check_ambiguity() {
    // Count frequencies of values in hidden array
    vector<int> cnt(5, 0);
    for (int x : a) cnt[x]++;

    bool possible_triangle = false;
    // Iterate all possible sorted triplets of values (u, v, w)
    for (int u = 1; u <= 4; u++) {
        for (int v = u; v <= 4; v++) {
            for (int w = v; w <= 4; w++) {
                if (is_triangle(u, v, w)) {
                    // Check if hidden array has enough of these values to form this
    triangle
                    int need[5] = {0};
                    need[u]++; need[v]++; need[w]++;
                    if (cnt[1] >= need[1] && cnt[2] >= need[2] &&
                        cnt[3] >= need[3] && cnt[4] >= need[4]) {
                        possible_triangle = true;
                    }
                }
            }
        }
    }

    // If no non-degenerate triangle can be formed, all queries return 0.
    if (!possible_triangle) return true;

    // Special case for N=3
    if (n == 3) {
        long long val = get_val(a[0], a[1], a[2]);
        if (val == 63) return true;
    }
```

```
        // Incorrect: assumes N>=4 and any triangle implies uniqueness.
        return false;
}
```

**Interactor excerpt: post-fix ambiguity check (statement-consistent)**

```cpp
bool check_ambiguity() {
    array<int, 5> cnt{};
    for (int x : a) cnt[x]++;

    auto pair_exists = [&](const array<int, 5>& c, int u, int v) -> bool {
        if (u == v) return c[u] >= 2;
        return c[u] >= 1 && c[v] >= 1;
    };

    auto equal_on_present_pairs = [&](int x, int y, const array<int, 5>& rest) -> bool
     {
        for (int u = 1; u <= 4; u++) {
            for (int v = u; v <= 4; v++) {
                if (!pair_exists(rest, u, v)) continue;
                if (get_val(x, u, v) != get_val(y, u, v)) return false;
            }
        }
        return true;
    };

    // Single-position replacement: change one x into y without changing any answers.
    for (int x = 1; x <= 4; x++) {
        if (cnt[x] == 0) continue;
        array<int, 5> rest = cnt;
        rest[x]--;
        for (int y = 1; y <= 4; y++) {
            if (y == x) continue;
            if (equal_on_present_pairs(x, y, rest)) return true;
        }
    }

    // Two-position swap: swap one x with one y.
    for (int x = 1; x <= 4; x++) {
        for (int y = x + 1; y <= 4; y++) {
            if (cnt[x] == 0 || cnt[y] == 0) continue;
            array<int, 5> rest = cnt;
            rest[x]--;
            rest[y]--;
            if (equal_on_present_pairs(x, y, rest)) return true;
        }
    }

    return false;
}
```

### D.7. Interactive I/O Templates

Interactive problems are sensitive to I/O synchronization and buffering. To reduce incidental failures unrelated to algorithmic reasoning, we provide optional language-specific I/O scaffolding for common languages. The templates emphasize explicit flushing after each query, graceful termination on EOF or invalid replies, and avoiding any non-protocol output on stdout.

**C++ interactive template**

```cpp
#include <bits/stdc++.h>
using namespace std;

// Interactive notes:
// - Print a query, then flush (endl is fine).
// - Read the judge reply immediately.
// - Many judges return -1 on invalid query; just exit.
// - Don't print anything else to stdout.

int main() {
  ios::sync_with_stdio(false);
  cin.tie(nullptr);

  // TODO: implement
  //
  // cout << "? " << x << "\n" << flush;
  // int r;
  // if (!(cin >> r)) return 0;
  // if (r == -1) return 0;
  //
  // cout << "! " << ans << "\n" << flush;

  return 0;
}
```

**Python interactive template**

```python
#!/usr/bin/env python3
import sys

input = sys.stdin.readline

# Interactive notes:
# - Print a query, then flush=True.
# - Read the judge reply immediately.
# - Many judges return -1 on invalid query; just return.
# - Don't print anything else to stdout.

def main() -> None:
    # TODO: implement
    #
    # print("?", x, flush=True)
    #     s = input()
    # if not s:
    #    return
    # s = s.strip()
    # if s == "-1": return
    #
    # print("!", ans, flush=True)
    return

if __name__ == "__main__":
    main()
```

**Java interactive template**

```java
import java.io.*;
import java.util.*;

// Interactive notes:
// - Print a query, then flush().
// - Read the judge reply immediately.
// - Many judges return -1 on invalid query; just return.
// - Don't print anything else to stdout.

public class Main {
    private static final BufferedReader br = new BufferedReader(new InputStreamReader(
    System.in));
    private static StringTokenizer st;
    private static final PrintWriter out = new PrintWriter(new BufferedWriter(new
    OutputStreamWriter(System.out)));

    private static String next() throws IOException {
        while (st == null || !st.hasMoreTokens()) {
            String line = br.readLine();
            if (line == null) System.exit(0);
            st = new StringTokenizer(line);
        }
        return st.nextToken();
    }

    private static int nextInt() throws IOException {
        return Integer.parseInt(next());
    }

    public static void main(String[] args) throws Exception {
        // TODO: implement
        //
        // out.println("? " + x);
        // out.flush();
        // Integer r = nextInt();
        // if (r == -1) return;
        //
        // out.println("! " + ans);
        // out.flush();
    }
}
```

**Go interactive template**

```go
package main

import (
  "bufio"
  "fmt"
  "os"
)

// Interactive notes:
// - Print a query, then Flush().
// - Read the judge reply immediately.
// - Many judges return -1 on invalid query; just exit.
// - Don't print anything else to stdout.
```

```go
func main() {
  in := bufio.NewReader(os.Stdin)
  out := bufio.NewWriter(os.Stdout)
  defer out.Flush()

  // TODO: implement
  //
  // fmt.Fprintln(out, "?", x)
  // out.Flush()
  // var r int
  // fmt.Fscan(in, &r)
  // if _, err := fmt.Fscan(in, &r); err != nil { return }
  //
  // fmt.Fprintln(out, "!", ans)
  // out.Flush()
}
```

### D.8. Prompt Templates for the Propose–Validate–Adjudicate Loop

This subsection lists the prompt templates used in our propose–validate–adjudicate loop (Section 3). We include them to improve auditability and transparency of the construction process.

**Prompt templates.**

- `generator.txt`: proposer prompt for generators.
- `interactor*.txt`: proposer prompts for interactors.
- `mutated_submissions.txt`: proposer prompt for mutated submissions.
- `decision*.txt`: adjudicator prompts for selecting candidates.

---

**Generator prompt: testlib.h generator**

```
You are an expert competitive programmer. Write a C++17 `testlib.h` generator that
    outputs EXACTLY ONE hidden test case for this interactive problem.

Constraints (one-shot):
- Do not rely on external tools, file access, or web browsing.
- Do not propose running commands or fetching additional information.
- Output a complete final C++17 program in one shot.

IMPORTANT (what this generator does):
- This generator is run OFFLINE to create `cases/*.in`.
- It outputs the HIDDEN INPUT that the interactor reads from `inf`.
  It is NOT an interaction transcript.
- The solver cannot see these `.in` files; only the interactor reads them.

Invocation (used in our pipeline):
- `gen <seed> [--key value / --key=value / -k value ...]`
- Use `registerGen(argc, argv, 1)` so `argv[1]` is the seed.
  - testlib's `rnd` becomes deterministic w.r.t. the seed; do NOT seed manually.

Requirements (strict):
1) Always include `#include "testlib.h"` FIRST, then `<bits/stdc++.h>`, and use `using
    namespace std;`.
2) Initialize with `registerGen(argc, argv, 1);`.
3) Use testlib randomness (`rnd.*`) only; do NOT use `rand()` and do NOT set seeds
    manually.
4) Parse args via `opt<>()` with sensible defaults; include `mode = opt<string>("mode
    ", "non")`.
```

---

```
5) Output exactly ONE test case to stdout in the EXACT hidden-input format the
     interactor expects for that mode.
6) Deterministic coverage:
   - small seeds (e.g., 1..3): hand-crafted corner cases (min, max, tricky).
   - other seeds: random stress + structured adversarial patterns.
7) Generate cases that stress query efficiency (hard instances near worst-case).

Common testlib pitfalls (avoid):
- Do NOT use `testlib::` namespace; symbols are global.
- When using `rnd.next(l, r)`, ensure `l` and `r` have the same type (both int or both
     long long).
- Do NOT call non-existent methods like `rnd.sample()`, `rnd.permutation()`, `rnd.
   nextf()`.

Output ONLY the C++ code, no markdown, no explanations.
```

## Interactor prompt

```
You are an expert competitive programmer. Write a NON-ADAPTIVE interactor (interactive
     judge) in C++17 using `testlib.h` for the given stdin/stdout interactive problem.

Constraints (one-shot):
- Do not rely on external tools, file access, or web browsing.
- Do not ask for more context.
- Output a complete final C++17 program in one shot.

EXECUTION MODEL (evaluation harness):
- Invoked as: `./interactor in.txt tout.txt`
- You MUST call `registerInteraction(argc, argv);`
  - `inf` reads the hidden input from `argv[1]` (`in.txt`, copied from `cases/*.in`).
  - `tout` writes logs to `argv[2]` (`tout.txt`).
  - `std::cin` reads the solver's outputs (pipe).
  - `std::cout` writes the interactor's replies to the solver (pipe).

HIDDEN INPUT RULE (non-adaptive):
- The hidden test case is FIXED and comes ONLY from `inf`.
- Do NOT generate new hidden data inside the interactor (no randomness, no sampling).
- The solver CANNOT read `in.txt`; any initial public parameters must be printed by
   the interactor to stdout as part of the protocol.

LOGGING CONTRACT (STRICT; parsed by the harness):
- Maintain TWO integer metrics (digits only):
  1) `queries`     = budget used `Q` (usually query count; if the problem uses "
     energy/cost", log the consumed cost instead).
  2) `query_limit` = budget limit `B` from the statement (integer; may depend on
     parameters read from `inf`).
- You MUST write BOTH of the following lines to `tout` on EVERY termination path (OK/
   WA/PE/RE/early exit):
  - `queries=<int>`
  - `query_limit=<int>`
- Lowercase keys exactly. No extra text on those lines.
- The runner uses regex `queries\s*=\s*(\d+)` and `query_limit\s*=\s*(\d+)`, and takes
     the LAST match.
  You may print multiple times defensively, but the final values must be correct.
- Use an `atexit(...)` guard PLUS a `finish(...)` wrapper that logs before calling `
   quitf(...)`.
- IMPORTANT: the harness may terminate stalled processes, which can bypass `atexit`.
  To avoid `queries=None` / `query_limit=None` in such failures, you MUST:
  - call `log_metrics()` once right after setting `query_limit` (before any blocking
     read from `std::cin`).
```

```
BUDGET ENFORCEMENT (important):
- Do NOT enforce the budget limit as WA/PE. Just count `queries` and report `
    query_limit`; the harness validates budget adherence using these counters.
- To prevent infinite loops / output spam, enforce a HARD CAP and terminate with PE if
     exceeded.
  - Recommended: `hard_cap = max(200000, 10 * query_limit)` (or a fixed large constant
     if needed).

ROBUST I/O (important for correctness + logging):
- Avoid `ouf.readInt()/ouf.readToken()/...` because malformed output may trigger
    internal `quitf` and skip your logging.
- Read solver output via `std::cin`, validate it manually, and decide verdict via your
     own `finish(...)`.
- After EVERY output to the solver (including initial parameters and every query reply
    ), flush.
  (`cout << endl;` is fine, or `cout.flush()`).
- When reading hidden input via testlib `inf`, DO NOT use `operator>>` (e.g. `inf >> x
    `) -- `InStream` does not implement it.
  Always use `inf.readInt() / inf.readLong() / inf.readToken()` (and validate ranges).

testlib notes:
- Always `#include "testlib.h"` first (then `<bits/stdc++.h>`).
- All testlib symbols are in the global namespace (do NOT use `testlib::`).
- Verdicts: `_ok` (0), `_wa` (1), `_pe` (2).

Minimal skeleton (fill in protocol-specific logic; print ONLY protocol outputs to
    stdout):

#include "testlib.h"
#include <bits/stdc++.h>
using namespace std;

static long long queries = 0;
static long long query_limit = 0;  // must be set based on the statement / inf
    parameters

static void log_metrics() {
    try {
        tout << "queries=" << queries << endl;
        tout << "query_limit=" << query_limit << endl;
    } catch (...) {
        // best-effort logging only
    }
}

static void finish(TResult verdict, const string& msg) {
    log_metrics();
    quitf(verdict, "%s", msg.c_str());
}

int main(int argc, char** argv) {
    registerInteraction(argc, argv);
    atexit(log_metrics);

    // TODO: read hidden input from `inf`
    // TODO: set query_limit (integer budget B)
    // TODO: implement protocol loop using std::cin/std::cout
}

Output ONLY the C++ code, no markdown, no explanations.
```

**Mutated-submission prompt**

```
You are an expert competitive programmer and evaluator engineer.
Your task is to write ONE mutated submission for a stdin/stdout interactive problem by
    lightly mutating a given baseline solver.

You will be given:
1) The problem statement and protocol specification (including the exact query/output
    formats and termination rules).
2) The query budget and any parameter constraints.
3) ONE baseline solver program that is Accepted by the official judge (source code).
Optionally, you may also be given a small set of representative Rejected submissions.
You will also be given a required category.

Goal:
This mutated submission is used as an internal adversarial stress test.
It is NOT for increasing the quantity of rejected submissions.
It is used to expose evaluator weaknesses that are hard to cover naturally in a small
    human verification pool:
1) The evaluator/interactor is too permissive: semantically incorrect behavior is
    mistakenly accepted.
2) Protocol/budget enforcement has gaps: invalid interaction is not consistently
    rejected.

Constraints (one-shot):
- Do not rely on external tools, file access, or web browsing.
- Do not ask for more context.
- Output a complete final program in one shot.

Output requirements:
- Output exactly ONE mutated submission program in the SAME language as the baseline
    solver.
- Keep changes minimal and localized. Do NOT rewrite the entire solver from scratch.
- The mutated submission should be plausible and close to correct behavior, not
    obviously sabotaged.

Required category:
You MUST follow the provided category, which is exactly one of the following.

Category A: SEMANTIC_WRONG_PROTOCOL_VALID
- The submission MUST follow the protocol and respect the query budget.
- The submission MUST flush after every query.
- The submission MUST not print anything to stdout outside the protocol.
- The submission MUST terminate normally.
- The submission MUST implement a subtle semantic bug so it is incorrect on some test
    cases.
  Prefer interaction-specific bugs such as:
  - incorrect state updates that break an invariant
  - early stopping with a plausible but incorrect final answer
  - misinterpreting judge feedback (e.g., flipping a binary reply, shifting a
    threshold)
  - dropping a necessary class of queries while keeping the protocol valid
  - small perturbations to the final output when answers are uniquely checkable

Category B: PROTOCOL_OR_BUDGET_WRONG
- The submission MUST compile and run, but MUST violate exactly one protocol or budget
    rule in a minimal, boundary way.
  Prefer fencepost violations that test completeness:
  - out-of-range query argument by 1
  - malformed token count or unsupported command word
  - exceeding the query limit by exactly 1
  - invalid termination behavior, such as printing after completion
- Do NOT combine multiple violations in one submission unless required by the protocol
```

```
      structure.

Output format (STRICT):
- Output ONLY code, no markdown and no explanations.
- Begin the program with a single-line comment header:
    // MUTATED_SUBMISSION | <CATEGORY> | <one short intent>
    where <CATEGORY> is exactly the provided category.
```

**Adjudicator prompt: select generator**

```
You are a judge selecting the best test case generator ('gen_cases.cpp') candidate.

You will be given:
1) The problem statement and meta.json
2) Hard constraints from 'INTERACTOR_GENERATOR_PLAN.md'
3) Candidate generator codes for a small set of models (exact names provided)

STRICT RULES:
- You MUST choose one model name from "Allowed models"
- You MUST output JSON only (single line), no explanations:
    {"pick": "<model_name>"}
- Do NOT invent or modify model names

SELECTION PRIORITY (in order):
1) Contract compliance: uses testlib generator pattern ('registerGen'), prints exactly
     one case to stdout,
    deterministic given seed, no hidden randomness.
2) Compatibility with the harness: supports the required mode(s) indicated by meta.
     json (e.g., '--mode non' or '--mode adp' when applicable).
3) Likely strength/diversity of generated cases and reasonable schema design.
4) Robustness and simplicity as tie-breaker.
```

**Adjudicator prompt: select interactor**

```
You are a judge selecting the best interactor candidate from multiple LLM-generated
    candidates.

You will be given:
1) The problem statement and meta.json
2) The selected generator code ('generator/gen_cases.cpp') to ensure schema coupling
3) Candidate interactor codes (non_adaptive.cpp and possibly adaptive.cpp)
4) Evaluation results from the harness for each candidate:
    - std-check: runs reference-correct solver programs in 'codes/std/*.cpp' if present
    . PASS means all cases are Accepted.
      NOTE: compile errors (CE) are ignored; if all std are CE (or the folder is empty)
    , std-check is SKIPPED.
    - std_wa-check: runs representative incorrect solver programs in 'codes/std_wa/*.
    cpp' if present. FAIL if any std_wa passes all cases.
    - If a check is marked SKIPPED, you MUST ignore that check (do not assume anything)
    .

STRICT RULES:
- You MUST choose one model name from "Allowed models"
- You MUST output JSON only (single line), no explanations:
    {"pick": "<model_name>"}
- Do NOT invent or modify model names

SELECTION PRIORITY (in order):
```

```
1) Correctness: if std-check exists, the interactor MUST PASS std-check.
2) Strength: if std_wa-check exists, the interactor MUST NOT allow any std_wa to pass
   all cases.
3) Query efficiency: prefer fewer queries if query stats are available.
4) Robustness and contract compliance (queries/query_limit logging, safe parsing,
   deterministic behavior).
```

**Adjudicator prompt: select solver and interactor**

```
You are a judge for selecting the best solver and interactor combination.
In the inputs, solver candidates are labeled as "std".

You will be given:
1. Tournament test results (PASS/FAIL for each std x interactor pair)
2. Valid model names for std (solver) and interactor
3. Code snippets of passing candidates (truncated)

STRICT RULES:
- You MUST choose a combination marked PASS in Tournament Results
- You MUST use exact model names from "Valid std models" and "Valid interactor models"
- Do NOT invent or modify model names

SELECTION PRIORITY (in order):
1. Correctness: must be PASS
2. Query efficiency: prefer solutions likely to use FEWER queries
3. Code clarity and robustness as tie-breaker

Output format (JSON only, single line, no explanation):
{"std": "<model_name>", "interactor": "<model_name>"}
```

### D.9. Model Performance by Category and Difficulty

We provide per-model category-wise results stratified by difficulty tiers (Easy/Medium/Hard). These tables complement Table 3 by reporting mean pass@1 and pass@5 for each category and difficulty tier, as well as overall.

*Table 15.* **Performance by category and difficulty: Gemini-3-Pro-Preview.**

| Category | Easy | | Medium | | Hard | | Overall | |
|---|---|---|---|---|---|---|---|---|
| | **pass@1** | **pass@5** | **pass@1** | **pass@5** | **pass@1** | **pass@5** | **pass@1** | **pass@5** |
| Graph | 0.713 | 0.870 | 0.467 | 0.711 | 0.220 | 0.500 | 0.475 | 0.705 |
| Search | 0.794 | 0.968 | 0.531 | 0.724 | 0.333 | 0.625 | 0.561 | 0.770 |
| Greedy | 0.738 | 0.938 | 0.390 | 0.667 | 0.382 | 0.636 | 0.470 | 0.725 |
| Bit | 0.883 | 1.000 | 0.645 | 0.727 | 0.255 | 0.364 | 0.613 | 0.711 |
| Data Structures | 0.843 | 1.000 | 0.416 | 0.686 | 0.364 | 0.727 | 0.487 | 0.750 |
| Math | 0.723 | 0.968 | 0.613 | 0.733 | 0.424 | 0.559 | 0.589 | 0.744 |
| Game | 0.764 | 1.000 | 0.311 | 0.611 | 0.467 | 0.667 | 0.480 | 0.743 |

*Table 16.* **Performance by category and difficulty: GPT-5.2.**

| Category | Easy | | Medium | | Hard | | Overall | |
|---|---|---|---|---|---|---|---|---|
| | **pass@1** | **pass@5** | **pass@1** | **pass@5** | **pass@1** | **pass@5** | **pass@1** | **pass@5** |
| Graph | 0.757 | 0.870 | 0.409 | 0.556 | 0.190 | 0.350 | 0.450 | 0.591 |
| Search | 0.735 | 0.968 | 0.545 | 0.690 | 0.308 | 0.417 | 0.547 | 0.708 |
| Greedy | 0.613 | 0.812 | 0.529 | 0.738 | 0.418 | 0.545 | 0.530 | 0.725 |
| Bit | 0.850 | 1.000 | 0.545 | 0.727 | 0.309 | 0.364 | 0.569 | 0.711 |
| Data Structures | 0.743 | 0.929 | 0.475 | 0.706 | 0.309 | 0.455 | 0.500 | 0.711 |
| Math | 0.735 | 0.903 | 0.580 | 0.700 | 0.376 | 0.559 | 0.563 | 0.712 |
| Game | 0.691 | 0.909 | 0.533 | 0.722 | 0.433 | 0.833 | 0.566 | 0.800 |

*Table 17.* **Performance by category and difficulty: DeepSeek-V3.2-Thinking.**

| Category | Easy | | Medium | | Hard | | Overall | |
|---|---|---|---|---|---|---|---|---|
| | **pass@1** | **pass@5** | **pass@1** | **pass@5** | **pass@1** | **pass@5** | **pass@1** | **pass@5** |
| Graph | 0.626 | 0.783 | 0.236 | 0.511 | 0.050 | 0.200 | 0.295 | 0.511 |
| Search | 0.639 | 0.806 | 0.286 | 0.466 | 0.083 | 0.208 | 0.340 | 0.504 |
| Greedy | 0.650 | 0.875 | 0.295 | 0.548 | 0.091 | 0.273 | 0.345 | 0.580 |
| Bit | 0.767 | 0.833 | 0.173 | 0.364 | 0.200 | 0.273 | 0.338 | 0.467 |
| Data Structures | 0.600 | 0.786 | 0.271 | 0.529 | 0.018 | 0.091 | 0.295 | 0.513 |
| Math | 0.555 | 0.774 | 0.313 | 0.483 | 0.094 | 0.176 | 0.314 | 0.472 |
| Game | 0.691 | 0.909 | 0.256 | 0.611 | 0.167 | 0.333 | 0.377 | 0.657 |

*Table 18.* **Performance by category and difficulty: Gemini-2.5-Pro.**

| Category | Easy | | Medium | | Hard | | Overall | |
|---|---|---|---|---|---|---|---|---|
| | pass@1 | pass@5 | pass@1 | pass@5 | pass@1 | pass@5 | pass@1 | pass@5 |
| Graph | 0.530 | 0.739 | 0.218 | 0.444 | 0.020 | 0.100 | 0.255 | 0.443 |
| Search | 0.587 | 0.742 | 0.266 | 0.448 | 0.067 | 0.167 | 0.312 | 0.469 |
| Greedy | 0.550 | 0.688 | 0.252 | 0.476 | 0.073 | 0.182 | 0.293 | 0.478 |
| Bit | 0.650 | 0.750 | 0.273 | 0.364 | 0.145 | 0.273 | 0.342 | 0.444 |
| Data Structures | 0.529 | 0.714 | 0.208 | 0.451 | 0.073 | 0.091 | 0.247 | 0.447 |
| Math | 0.503 | 0.710 | 0.320 | 0.483 | 0.106 | 0.235 | 0.307 | 0.472 |
| Game | 0.455 | 0.545 | 0.278 | 0.667 | 0.133 | 0.500 | 0.309 | 0.600 |

*Table 19.* **Performance by category and difficulty: GPT-OSS-120B.**

| Category | Easy | | Medium | | Hard | | Overall | |
|---|---|---|---|---|---|---|---|---|
| | pass@1 | pass@5 | pass@1 | pass@5 | pass@1 | pass@5 | pass@1 | pass@5 |
| Graph | 0.522 | 0.783 | 0.151 | 0.333 | 0.010 | 0.050 | 0.216 | 0.386 |
| Search | 0.535 | 0.677 | 0.197 | 0.345 | 0.100 | 0.208 | 0.269 | 0.407 |
| Greedy | 0.525 | 0.750 | 0.171 | 0.310 | 0.000 | 0.000 | 0.226 | 0.362 |
| Bit | 0.583 | 0.917 | 0.227 | 0.409 | 0.055 | 0.273 | 0.280 | 0.511 |
| Data Structures | 0.414 | 0.643 | 0.157 | 0.275 | 0.018 | 0.091 | 0.184 | 0.316 |
| Math | 0.432 | 0.710 | 0.260 | 0.400 | 0.082 | 0.206 | 0.254 | 0.424 |
| Game | 0.564 | 0.727 | 0.156 | 0.333 | 0.200 | 0.333 | 0.291 | 0.457 |

*Table 20.* **Performance by category and difficulty: Claude-Opus-4.5.**

| Category | Easy | | Medium | | Hard | | Overall | |
|---|---|---|---|---|---|---|---|---|
| | pass@1 | pass@5 | pass@1 | pass@5 | pass@1 | pass@5 | pass@1 | pass@5 |
| Graph | 0.296 | 0.565 | 0.120 | 0.222 | 0.000 | 0.000 | 0.139 | 0.261 |
| Search | 0.271 | 0.484 | 0.148 | 0.259 | 0.008 | 0.042 | 0.152 | 0.274 |
| Greedy | 0.375 | 0.688 | 0.095 | 0.238 | 0.018 | 0.091 | 0.148 | 0.319 |
| Bit | 0.317 | 0.667 | 0.045 | 0.182 | 0.055 | 0.091 | 0.120 | 0.289 |
| Data Structures | 0.343 | 0.714 | 0.098 | 0.255 | 0.000 | 0.000 | 0.129 | 0.303 |
| Math | 0.258 | 0.484 | 0.193 | 0.350 | 0.029 | 0.059 | 0.165 | 0.304 |
| Game | 0.345 | 0.636 | 0.078 | 0.222 | 0.000 | 0.000 | 0.149 | 0.314 |

*Table 21.* **Performance by category and difficulty: Gemini-2.5-Flash.**

| Category | Easy | | Medium | | Hard | | Overall | |
|---|---|---|---|---|---|---|---|---|
| | **pass@1** | **pass@5** | **pass@1** | **pass@5** | **pass@1** | **pass@5** | **pass@1** | **pass@5** |
| Graph | 0.287 | 0.478 | 0.076 | 0.156 | 0.000 | 0.000 | 0.114 | 0.205 |
| Search | 0.310 | 0.484 | 0.162 | 0.328 | 0.008 | 0.042 | 0.170 | 0.310 |
| Greedy | 0.312 | 0.688 | 0.157 | 0.357 | 0.018 | 0.091 | 0.171 | 0.391 |
| Bit | 0.400 | 0.667 | 0.082 | 0.182 | 0.018 | 0.091 | 0.151 | 0.289 |
| Data Structures | 0.186 | 0.429 | 0.141 | 0.333 | 0.000 | 0.000 | 0.129 | 0.303 |
| Math | 0.271 | 0.548 | 0.200 | 0.333 | 0.012 | 0.059 | 0.166 | 0.312 |
| Game | 0.382 | 0.818 | 0.122 | 0.333 | 0.000 | 0.000 | 0.183 | 0.429 |

*Table 22.* **Performance by category and difficulty: GPT-OSS-20B.**

| Category | Easy | | Medium | | Hard | | Overall | |
|---|---|---|---|---|---|---|---|---|
| | **pass@1** | **pass@5** | **pass@1** | **pass@5** | **pass@1** | **pass@5** | **pass@1** | **pass@5** |
| Graph | 0.330 | 0.609 | 0.067 | 0.178 | 0.000 | 0.000 | 0.120 | 0.250 |
| Search | 0.310 | 0.516 | 0.090 | 0.207 | 0.025 | 0.083 | 0.136 | 0.265 |
| Greedy | 0.425 | 0.750 | 0.076 | 0.238 | 0.000 | 0.000 | 0.145 | 0.319 |
| Bit | 0.233 | 0.500 | 0.055 | 0.136 | 0.036 | 0.182 | 0.098 | 0.244 |
| Data Structures | 0.200 | 0.571 | 0.047 | 0.157 | 0.000 | 0.000 | 0.068 | 0.211 |
| Math | 0.239 | 0.516 | 0.157 | 0.317 | 0.024 | 0.088 | 0.141 | 0.304 |
| Game | 0.364 | 0.727 | 0.067 | 0.222 | 0.000 | 0.000 | 0.149 | 0.343 |

*Table 23.* **Performance by category and difficulty: Qwen3-30B-A3B-Thinking.**

| Category | Easy | | Medium | | Hard | | Overall | |
|---|---|---|---|---|---|---|---|---|
| | **pass@1** | **pass@5** | **pass@1** | **pass@5** | **pass@1** | **pass@5** | **pass@1** | **pass@5** |
| Graph | 0.174 | 0.348 | 0.107 | 0.178 | 0.010 | 0.050 | 0.102 | 0.193 |
| Search | 0.232 | 0.452 | 0.097 | 0.241 | 0.008 | 0.042 | 0.115 | 0.257 |
| Greedy | 0.200 | 0.625 | 0.167 | 0.286 | 0.036 | 0.182 | 0.154 | 0.348 |
| Bit | 0.217 | 0.500 | 0.055 | 0.091 | 0.000 | 0.000 | 0.084 | 0.178 |
| Data Structures | 0.171 | 0.429 | 0.094 | 0.196 | 0.000 | 0.000 | 0.095 | 0.211 |
| Math | 0.071 | 0.226 | 0.123 | 0.267 | 0.018 | 0.088 | 0.082 | 0.208 |
| Game | 0.218 | 0.636 | 0.122 | 0.167 | 0.033 | 0.167 | 0.137 | 0.314 |

*Table 24.* **Performance by category and difficulty: Qwen3-32B.**

| Category | Easy | | Medium | | Hard | | Overall | |
|---|---|---|---|---|---|---|---|---|
| | pass@1 | pass@5 | pass@1 | pass@5 | pass@1 | pass@5 | pass@1 | pass@5 |
| Graph | 0.270 | 0.435 | 0.076 | 0.111 | 0.010 | 0.050 | 0.111 | 0.182 |
| Search | 0.245 | 0.419 | 0.062 | 0.138 | 0.008 | 0.042 | 0.101 | 0.195 |
| Greedy | 0.300 | 0.625 | 0.067 | 0.119 | 0.018 | 0.091 | 0.113 | 0.232 |
| Bit | 0.267 | 0.417 | 0.055 | 0.091 | 0.000 | 0.000 | 0.098 | 0.156 |
| Data Structures | 0.271 | 0.500 | 0.035 | 0.078 | 0.000 | 0.000 | 0.074 | 0.145 |
| Math | 0.065 | 0.194 | 0.077 | 0.133 | 0.012 | 0.059 | 0.056 | 0.128 |
| Game | 0.327 | 0.636 | 0.033 | 0.056 | 0.000 | 0.000 | 0.120 | 0.229 |

*Table 25.* **Performance by category and difficulty: Qwen3-14B.**

| Category | Easy | | Medium | | Hard | | Overall | |
|---|---|---|---|---|---|---|---|---|
| | pass@1 | pass@5 | pass@1 | pass@5 | pass@1 | pass@5 | pass@1 | pass@5 |
| Graph | 0.139 | 0.261 | 0.049 | 0.133 | 0.000 | 0.000 | 0.061 | 0.136 |
| Search | 0.187 | 0.290 | 0.041 | 0.086 | 0.000 | 0.000 | 0.073 | 0.124 |
| Greedy | 0.212 | 0.438 | 0.029 | 0.095 | 0.000 | 0.000 | 0.067 | 0.159 |
| Bit | 0.133 | 0.167 | 0.027 | 0.045 | 0.000 | 0.000 | 0.049 | 0.067 |
| Data Structures | 0.214 | 0.429 | 0.020 | 0.078 | 0.000 | 0.000 | 0.053 | 0.132 |
| Math | 0.032 | 0.097 | 0.067 | 0.167 | 0.000 | 0.000 | 0.040 | 0.104 |
| Game | 0.164 | 0.455 | 0.044 | 0.111 | 0.000 | 0.000 | 0.074 | 0.200 |

*Table 26.* **Performance by category and difficulty: DeepSeek-R1-Distill-Llama-70B.**

| Category | Easy | | Medium | | Hard | | Overall | |
|---|---|---|---|---|---|---|---|---|
| | pass@1 | pass@5 | pass@1 | pass@5 | pass@1 | pass@5 | pass@1 | pass@5 |
| Graph | 0.070 | 0.217 | 0.040 | 0.111 | 0.000 | 0.000 | 0.039 | 0.114 |
| Search | 0.181 | 0.323 | 0.045 | 0.086 | 0.000 | 0.000 | 0.073 | 0.133 |
| Greedy | 0.162 | 0.438 | 0.024 | 0.071 | 0.000 | 0.000 | 0.052 | 0.145 |
| Bit | 0.017 | 0.083 | 0.027 | 0.045 | 0.000 | 0.000 | 0.018 | 0.044 |
| Data Structures | 0.157 | 0.286 | 0.012 | 0.059 | 0.000 | 0.000 | 0.037 | 0.092 |
| Math | 0.039 | 0.065 | 0.057 | 0.100 | 0.000 | 0.000 | 0.037 | 0.064 |
| Game | 0.109 | 0.364 | 0.044 | 0.111 | 0.000 | 0.000 | 0.057 | 0.171 |

*Table 27.* **Performance by category and difficulty: DeepSeek-V3.2.**

| Category | Easy | | Medium | | Hard | | Overall | |
|---|---|---|---|---|---|---|---|---|
| | **pass@1** | **pass@5** | **pass@1** | **pass@5** | **pass@1** | **pass@5** | **pass@1** | **pass@5** |
| Graph | 0.304 | 0.478 | 0.107 | 0.244 | 0.020 | 0.050 | 0.139 | 0.261 |
| Search | 0.323 | 0.548 | 0.138 | 0.224 | 0.025 | 0.083 | 0.165 | 0.283 |
| Greedy | 0.362 | 0.562 | 0.124 | 0.262 | 0.000 | 0.000 | 0.159 | 0.290 |
| Bit | 0.350 | 0.500 | 0.082 | 0.182 | 0.018 | 0.091 | 0.138 | 0.244 |
| Data Structures | 0.329 | 0.643 | 0.051 | 0.176 | 0.000 | 0.000 | 0.095 | 0.237 |
| Math | 0.161 | 0.323 | 0.173 | 0.250 | 0.029 | 0.059 | 0.131 | 0.216 |
| Game | 0.273 | 0.455 | 0.089 | 0.167 | 0.000 | 0.000 | 0.131 | 0.229 |

*Table 28.* **Performance by category and difficulty: GPT-4.1.**

| Category | Easy | | Medium | | Hard | | Overall | |
|---|---|---|---|---|---|---|---|---|
| | **pass@1** | **pass@5** | **pass@1** | **pass@5** | **pass@1** | **pass@5** | **pass@1** | **pass@5** |
| Graph | 0.130 | 0.391 | 0.044 | 0.111 | 0.000 | 0.000 | 0.057 | 0.159 |
| Search | 0.187 | 0.290 | 0.024 | 0.069 | 0.000 | 0.000 | 0.064 | 0.115 |
| Greedy | 0.212 | 0.438 | 0.048 | 0.119 | 0.000 | 0.000 | 0.078 | 0.174 |
| Bit | 0.133 | 0.250 | 0.000 | 0.000 | 0.000 | 0.000 | 0.036 | 0.067 |
| Data Structures | 0.129 | 0.286 | 0.043 | 0.118 | 0.000 | 0.000 | 0.053 | 0.132 |
| Math | 0.090 | 0.258 | 0.037 | 0.083 | 0.006 | 0.029 | 0.042 | 0.112 |
| Game | 0.145 | 0.545 | 0.011 | 0.056 | 0.033 | 0.167 | 0.057 | 0.229 |

*Table 29.* **Performance by category and difficulty: Qwen3-32B-NonThinking.**

| Category | Easy | | Medium | | Hard | | Overall | |
|---|---|---|---|---|---|---|---|---|
| | **pass@1** | **pass@5** | **pass@1** | **pass@5** | **pass@1** | **pass@5** | **pass@1** | **pass@5** |
| Graph | 0.026 | 0.087 | 0.013 | 0.022 | 0.000 | 0.000 | 0.014 | 0.034 |
| Search | 0.039 | 0.129 | 0.003 | 0.017 | 0.000 | 0.000 | 0.012 | 0.044 |
| Greedy | 0.050 | 0.188 | 0.005 | 0.024 | 0.000 | 0.000 | 0.014 | 0.058 |
| Bit | 0.000 | 0.000 | 0.045 | 0.045 | 0.000 | 0.000 | 0.022 | 0.022 |
| Data Structures | 0.043 | 0.143 | 0.000 | 0.000 | 0.000 | 0.000 | 0.008 | 0.026 |
| Math | 0.026 | 0.065 | 0.010 | 0.017 | 0.000 | 0.000 | 0.011 | 0.024 |
| Game | 0.055 | 0.091 | 0.000 | 0.000 | 0.000 | 0.000 | 0.017 | 0.029 |

*Table 30.* **Performance by category and difficulty: Qwen3-14B-NonThinking.**

| Category | Easy | | Medium | | Hard | | Overall | |
|---|---|---|---|---|---|---|---|---|
| | **pass@1** | **pass@5** | **pass@1** | **pass@5** | **pass@1** | **pass@5** | **pass@1** | **pass@5** |
| Graph | 0.026 | 0.087 | 0.027 | 0.044 | 0.000 | 0.000 | 0.020 | 0.045 |
| Search | 0.032 | 0.129 | 0.007 | 0.034 | 0.000 | 0.000 | 0.012 | 0.053 |
| Greedy | 0.050 | 0.125 | 0.005 | 0.024 | 0.000 | 0.000 | 0.014 | 0.043 |
| Bit | 0.000 | 0.000 | 0.000 | 0.000 | 0.000 | 0.000 | 0.000 | 0.000 |
| Data Structures | 0.043 | 0.143 | 0.004 | 0.020 | 0.000 | 0.000 | 0.011 | 0.039 |
| Math | 0.013 | 0.032 | 0.003 | 0.017 | 0.000 | 0.000 | 0.005 | 0.016 |
| Game | 0.055 | 0.182 | 0.000 | 0.000 | 0.000 | 0.000 | 0.017 | 0.057 |

