# OpenReview forum: "InteractBench: Benchmarking LLMs on Competitive Programming under Unrevealed Information"
_ICML.cc/2026/Conference — ICML 2026 regular_

### Official Review · Reviewer_TdgR · 2026-03-05

**Soundness:** 3
**Presentation:** 2
**Significance:** 3
**Originality:** 2
**Overall Recommendation:** 4
**Confidence:** 3

**Summary:**

This paper introduces InteractBench, a novel benchmark designed to evaluate LLMs on competitive programming tasks involving unrevealed information. Through an extensive evaluation of state-of-the-art models, the authors highlight significant performance gaps in current LLMs when handling these tasks. To provide deeper insight, the study includes a fine-grained failure taxonomy that identifies and categorizes the root causes of the failures.

**Compliance With Llm Reviewing Policy:**

Affirmed.

**Final Justification:**

I appreciate the authors' effort in providing a thorough reply. Since my original concerns have been fully resolved, I have raised my score from 3 to 4.

**Key Questions For Authors:**

1. Could the authors provide a comparative evaluation to isolate the impact of unrevealed information on model performance? (e.g., comparing the pass rate of models with access to complete input data versus those using only the interactor).

2. Would the authors be able to provide results for a representative agentic baseline (e.g., an LLM with agent scaffolding or multi-turn refinement) to demonstrate how such approaches impact the pass rate on InteractBench?

**Limitations:**

While an impact statement is included, the manuscript lacks a formal discussion of limitations. The authors may include a dedicated *Limitations* section—either in the main text or the appendix—to address the constraints of the benchmark and the scope of the evaluation.

**Strengths And Weaknesses:**

Strength:

1. The work addresses a significant research gap by evaluating LLM performance in competitive programming under conditions of unrevealed information. This is a timely contribution as the field moves toward more complex, interactive problem-solving benchmarks.

2. Although the benchmark is largely LLM-generated, the authors have implemented a rigorous validation pipeline that ensures high data reliability and quality.

3. The paper provides a comprehensive evaluation of state-of-the-art LLMs, effectively highlighting the current limitations of these models when faced with information-unrevealed programming tasks.

Weaknesses:

1. The current *Evaluation Results* section is somewhat descriptive, primarily rephrasing raw statistical data without offering sufficient insight. The paper’s impact would be significantly bolstered by a more technical discussion on the mechanisms behind these results—for instance, a case study of why reasoning models specifically outperform non-reasoning counterparts—and a discussion on how these findings can inform the development of future LLMs.

2. Given that the benchmark focuses on unrevealed information, the study would benefit from a controlled comparative experiment. Specifically, comparing LLM pass rates when provided with complete input data versus the current interactor-only setup would help quantify the exact impact of information restriction on performance.

3. The current evaluation methodology relies on zero-shot prompts and single-turn LLM queries. However, the failure-mode decomposition indicates that a large portion of errors (e.g., PE and CE) might be mitigated through multi-turn refinement of agentic frameworks. Including an evaluation of agentic approaches is crucial, as it could fundamentally shift the interpretation of the results and the reported failure rates.

4. The presentation of formal concepts could be more precise. I recommend adding rigorous symbolic definitions to the "Problem Definitions" and "Metrics" sections. Furthermore, key details regarding the difficulty levels and categories of the programming problems should be integrated into the main text rather than relegated to the Appendix to ensure a smoother narrative flow and better accessibility for the reader.

---

> ### Author Rebuttal · Authors · 2026-03-31
>
> We thank the reviewer for the thorough evaluation and for recognizing that InteractBench addresses ''a significant research gap'' with ''a rigorous validation pipeline'' and ''a comprehensive evaluation''. The call for deeper mechanistic analysis is especially well taken, and we have conducted a controlled batch-vs.-interactive comparison, iterative refinement and prompting studies, and code-level case analyses in response. We address each concern below and hope these additions adequately resolve the remaining concerns; we respectfully ask the reviewer to reconsider the assessment.
>
> > **W1: Results lack mechanistic explanations and implications for future LLMs**
>
> **A:** This is an insightful suggestion. We have strengthened the Results section with both aggregate analysis and code-level case studies. The failure distribution reveals a consistent depth gradient: frontier models concentrate failures in WA and QLE (later interaction stages), while weaker models fail at PE and IDLE — for instance, on one problem GPT-5.2 exceeds the query budget by just 0.4% while a weaker model deadlocks with zero queries; both "fail," yet the taxonomy captures the stark gap that pass/fail misses.
>
> To illustrate, we compare DeepSeek-V3.2-Thinking and DeepSeek-V3.2 on an ICPC problem requiring interactive identification under a strict query budget. The non-reasoning variant fails in 8/10 samples by violating the bound directly; the reasoning variant succeeds in all 10 by first decomposing the task into subgoals, then designing a bounded query strategy. This form of **task decomposition** involves deciding what information to acquire, a challenge absent in batch evaluation where all input is given upfront.
>
> Taken together, the aggregate gradient and the case study above point to two directions: interactive I/O patterns appear underrepresented in training data (given persistent PE/IDLE in frontier models), and QLE concentrating in otherwise-capable models suggests that query efficiency is a distinct capability dimension from algorithmic correctness. Controlled experiments (W2, W3) and cross-language breakdowns further extend this analysis.
>
> > **W2, Q1: Lack of batch vs interactive controlled experiment**
>
> **A:** We fully agree — this controlled comparison goes to the heart of our central claim, and we are grateful the reviewer raised it. We have conducted a controlled experiment on 150 problems where the only difference is that the hidden test case is revealed through standard input. Revealing hidden information consistently improves pass@1 by 0.037–0.256 across all five models; unrevealed information is itself a substantial source of difficulty beyond the algorithmic core. Failure composition shifts correspondingly: interaction-specific errors (PE, QLE) largely disappear in the batch setting, while WA rises to dominate. Full results appear in Table 3 ([link](https://bit.ly/4s5lw6E)); a comprehensive discussion is provided in our response to **Reviewer b2hV (W1)**.
>
> > **W3, Q2: Lack of agentic baseline**
>
> **A:** This is an important concern. We have conducted both iterative refinement (refine@5) and few-shot prompting experiments across all 322 problems and five models. Surface-level errors (CE, PE) are indeed partially recoverable, but WA persists as the dominant failure mode and the cumulative gain is strongly capability-dependent (GPT-5.2: +0.146 vs. Qwen3-14B-NonThinking: +0.010). The two interventions are complementary: few-shot helps weaker models with protocol compliance, while iterative refinement benefits stronger models. Among all strategies, independent resampling (pass@5) gives the strongest overall gains. Full breakdowns appear in Tables 1 and 2 ([link](https://bit.ly/4s5lw6E)); a comprehensive discussion is provided in our response to **Reviewer b2hV (W2, W3)**.
>
> > **W4: Formal definitions and presentation**
>
> **A:** We are grateful for this concrete suggestion and have followed both recommendations.
>
> In Problem Definitions, the acceptance criterion is now explicit: a run with transcript $\tau$, query count $T$, and final answer $a$ is Accepted iff (1) $\tau$ conforms to protocol $\Pi$, (2) $T \le B$, and (3) $a$ is correct given hidden input $x$ and $\tau$.
>
> In Metrics, the failure taxonomy is formalized as a priority-ordered classification. If harness status $h \ne \mathrm{OK}$ (i.e., CE/RE/TLE/MLE/IDLE), the label is $h$ directly; otherwise the label follows the first violated acceptance condition: PE if the protocol is violated, QLE if $T > B$, WA if the answer is incorrect. The difficulty distribution and algorithm-category breakdown have also been moved into the main text.
>
> > **L1: Lack of formal Limitations discussion**
>
> **A:** We agree this improves the manuscript and have added a dedicated Limitations section consolidating the scope boundaries discussed throughout the paper, with references to the additional rebuttal experiments.

---

> > ### Author Rebuttal · Reviewer_TdgR · 2026-04-01
> >
> > I appreciate the authors' effort in providing a thorough reply. Since my original concerns have been fully resolved, I have raised my score from 3 to 4.

---

> > > ### Author Response · Authors · 2026-04-08
> > >
> > > We sincerely thank the reviewer for the thoughtful follow-up and for raising the score after our rebuttal. We greatly appreciate your recognition that the additional analyses and clarifications have fully addressed the original concerns, and we have incorporated the corresponding revisions into the manuscript. We would be grateful for your further support.

---

### Official Review · Reviewer_RyoL · 2026-03-12

**Soundness:** 3
**Presentation:** 3
**Significance:** 3
**Originality:** 3
**Overall Recommendation:** 5
**Confidence:** 4

**Summary:**

This paper proposes the InteractBench, an interactive benchmark for evaluating LLMs’ algorithmic reasoning abilities. This benchmark provides upgrades from static benchmarks like LiveCodeBench. Authors clearly state limitations of existing benchmarks, and show the importance of InteractBench. On this benchmark, authors evaluate 16 model configurations and provide findings. They show that even the most advanced reasoning models achieve limited success on interactive problems. They also propose the fine-grained error taxonomy for better understanding limitations of existing models.

**Compliance With Llm Reviewing Policy:**

Affirmed.

**Final Justification:**

Authors addressed all concerns raised by me, and I am satisfied with the detailed experiments and explanations they provided.

**Key Questions For Authors:**

- Have you considered evaluating models in a turn-by-turn agentic setting, rather than generating a single self-contained program? How might the performance differ?
- Is the propose-validate-adjudicate pipeline fully automated enough to scale to thousands of problems?, if yes, then why did we not scale the dataset?

**Limitations:**

Yes

**Strengths And Weaknesses:**

Strengths:

- Creating interactive benchmarks in code generating/solving settings is novel and valuable.
- Coverage of benchmark in terms of different algorithm categories is good despite being only 322 instances.
- Benchmark construction method is innovative, and includes human-in-loop which is important for quality.
- Choice of model configurations is comprehensive, and paper is well-written.

Weaknesses:

- Error taxonomy and final performance are good metrics, however, I did not see any discussions regarding interactive capabilities of each LLMs. Mainly, how good they are at doing such interactions (maybe some partial accuracy or something) although they fail on final tasks. Such explicit evaluations can be important and give more clarity on the utility of benchmark.
- Currently the major programming language covered in benchmark is only C++, I think based on this, I am not sure we should make generalizable claims about model behaviours. This needs to be assessed thoroughly across programming languages.
- The paper mentions that I/O templates can have a non-monotonic effect on performance across different languages. It would be helpful to see more analysis on how sensitive the models are to the prompt instructions regarding interaction protocols.

---

> ### Author Rebuttal · Authors · 2026-03-31
>
> We thank the reviewer for the constructive feedback and for recognizing that interactive benchmarking is ''novel and valuable'', that the benchmark construction method is ''innovative'', and that the model configurations are ''comprehensive''. We hope the additional analyses below help address the remaining concerns.
>
> > **W1: Fine-grained evaluation of interaction capability**
>
> **A:** This is an excellent point. We agree that interaction capability deserves characterization beyond aggregate pass/fail. Our error taxonomy already captures this: models whose failures concentrate in WA/QLE have progressed further through the protocol than those failing at CE/PE/IDLE. In Exp#3, frontier models indeed show this pattern, and we have made the ordering more explicit in the revision.
>
> As a complementary continuous measure, we have computed the partial test-case pass rate among failed submissions. GPT-5.2 achieves 0.317 while Qwen3-14B-NonThinking reaches only 0.096, mirroring the pass@1 ordering and suggesting that stronger models retain more partial correctness even when they fail.
>
> > **W2: Cross-language generalizability**
>
> **A:** Thank you for raising this concern. Appendix D.4 already reports evaluation across C++, Python, Java, and Go with and without I/O templates; we have since extended it to 5 models and computed the full failure-type breakdown.
>
> The main conclusions hold across languages:
>
> 1. **Model ranking is broadly preserved.** Across all 8 configurations (4 languages × 2 template settings), no rank inversions occur in pass@1 ordering.
>
> 2. **WA remains the dominant bottleneck once language-specific implementation failures are controlled.** In settings not dominated by CE, WA is consistently the most prevalent failure type, suggesting that the main remaining bottleneck is algorithmic reasoning under partial observability rather than language-specific implementation artifacts. For example, WA accounts for 28–30% of all problems for GPT-5.2 in C++ and Python, and over 58% for Qwen3-14B in the same settings.
>
> 3. **QLE appears more consistently in stronger models.** GPT-5.2 shows QLE rates of 4–5% across languages, while Qwen3-14B-NonThinking stays at 0.0–6.0%, mirroring the C++ finding.
>
> 4. **The reasoning boost persists.** DeepSeek-V3.2-Thinking outperforms DeepSeek-V3.2 on pass@1 in all 8 configurations, with relative gains ranging from 2.3× to 5.2×.
>
> 5. **Cross-language variation stems from non-interaction factors.** CE is elevated in Go and Java; IDLE is elevated in Python without templates. C++ exhibits consistently low CE and IDLE rates across all five models, supporting our choice of it as the default setting.
>
> The complete cross-language breakdown is in Table 4 ([link](https://bit.ly/4s5lw6E)). We hope these results adequately address the generalizability concern.
>
> > **W3: Prompt sensitivity analysis**
>
> **A:** We appreciate this suggestion. Template effects indeed deserve mechanistic analysis. We have decomposed the effect by failure type across 5 models × 4 languages.
>
> 1. **Templates mainly reduce IDLE and often shift failures toward WA, rather than resolving the underlying reasoning errors.** The strongest effect appears in Python, where models often deadlock on I/O synchronization without a template. For example, DeepSeek-V3.2 on Python sees IDLE drop from 39.6% to 5.0% with a template, while WA rises from 35.9% to 59.1%, exposing algorithmic errors that were previously masked by deadlocks.
>
> 2. **The net effect is non-monotonic.** Templates reduce I/O failures but may constrain solution strategies in some cases. For DeepSeek-V3.2-Thinking on Python, the gain is limited, likely because its no-template IDLE rate is already low.
>
> 3. **C++ is the least template-sensitive setting.** Across all five models, the absolute pass@1 difference between template and no-template is smallest for C++ (≤0.030), compared with Python (up to 0.071) and Go (up to 0.044).
>
> The full failure-type decomposition is also in Table 4 ([link](https://bit.ly/4s5lw6E)).
>
> > **Q1: Agentic turn-by-turn evaluation**
>
> **A:** Thank you for this suggestion. This concern is shared across all three reviewers, and we have conducted iterative refinement (refine@5) and few-shot prompting experiments across all 322 problems and five models. Key finding: CE and PE are partially recoverable, but WA persists and the cumulative gain is strongly capability-dependent (GPT-5.2: +0.146 vs. Qwen3-14B-NonThinking: +0.010). A detailed treatment is given in our response to **Reviewer b2hV (W2, W3)**.
>
> > **Q2: Pipeline scalability**
>
> **A:** This is a fair question. Interactive problems are a small minority on major competitive-programming platforms, and our dataset already covers a significant portion of eligible candidates. The scaling bottleneck is therefore source-material scarcity rather than pipeline capacity. Synthesizing novel interactive problems is a promising direction to overcome this ceiling.

---

> > ### Author Rebuttal · Reviewer_RyoL · 2026-04-03
> >
> > Thank you for detailed rebuttal and resolving my concerns. I suggest authors to add all the discussions and updated results to the revised version.
> >
> > I improve my score from 4 to 5.

---

> > > ### Author Response · Authors · 2026-04-08
> > >
> > > We sincerely thank the reviewer for the very encouraging feedback and for raising the score after reading our rebuttal. We truly appreciate your recognition that our response has resolved the concerns, and we have carefully incorporated all additional discussions and updated results into the revised manuscript; we would be especially grateful for your continued support.

---

### Official Review · Reviewer_b2hV · 2026-03-26

**Soundness:** 3
**Presentation:** 4
**Significance:** 3
**Originality:** 3
**Overall Recommendation:** 4
**Confidence:** 4

**Summary:**

InteractBench introduces 322 problems to evaluate LLMs on interactive competitive programming, where models must dynamically acquire hidden information through multi-round queries rather than reading static inputs. It provides a self-contained offline harness using local interactors, completely bypassing the need for external online judges. Testing across 16 models reveals a steep performance drop on hard tasks and shows that models frequently fail due to interaction-specific issues (protocol violations, deadlocks, budget overruns) alongside standard algorithmic logic errors.

**Compliance With Llm Reviewing Policy:**

Affirmed.

**Final Justification:**

I thank the authors for their rebuttal and will maintain my positive score.

**Key Questions For Authors:**

1. What is the sensitivity of pass@k estimates to the number of test cases per task? Have you experimented with varying suite sizes (e.g., 5 vs. 10)?
2. How do you mitigate the circularity risk of proposer models being evaluated on their own interactors? Would a leave-one-out protocol (where each model is evaluated only on interactors it did not help construct) be feasible?

**Limitations:**

yes

**Strengths And Weaknesses:**

## Strengths
1. **Well-motivated Gap:** while interactive code generation problems have appeared sporadically in prior benchmarks (e.g., 21 tasks in LiveCodeBench Pro, 23 tasks in LiveOIBench, 17 tasks in CodeElo), InteractBench is the first to systematically study this concept at scale.

2. **Robust Evaluation:** The self-contained offline harness with local interactors is a substantial engineering contribution. The propose–validate–adjudicate construction pipeline, mutation-based stress probes, and post-hoc online agreement audits, providing strong guarantees on evaluator fidelity. The inclusion of adaptive interactors for tasks permitting history-dependent judge behavior further strengthens the benchmark's rigor.

3. **Actionable Failure Modes:** The decomposition of failure modes (PE/QLE/IDLE/CE/RE/TLE/MLE/WA) goes well beyond standard pass@k score reporting. This yields genuinely useful insights. For instance, the finding that frontier models disproportionately suffer from QLE (query-budget overruns, e.g., 11.9% for Gemini-3-Pro-Preview), while weaker models fail earlier on protocol compliance suggests that query efficiency is a distinct capability dimension from algorithmic correctness.

4. **Comprehensive Experiment:** The evaluation covers 16 models with task stratification by difficulty and category, within-family reasoning vs. non-reasoning comparisons (e.g., DeepSeek-V3.2 vs. DeepSeek-V3.2-Thinking), temporal contamination diagnostics via time splits, cross-language robustness analysis, and I/O template ablations. This thoroughness increases the reliability of the reported findings.

## Weaknesses
1. **Confounding between interaction and difficulty.** The paper demonstrates an "interaction gap" but fails to isolate whether models struggle with dynamic information acquisition or if interactive problems are simply harder. A controlled experiment comparing batch (full input upfront) and interactive (queried input) variants of the exact same algorithmic core is needed to validate this central claim.

2. **Lack of agentic evaluation.** Given that many failures stem from shallow implementation issues (e.g., format errors, missing flushes), testing an agentic harness (e.g. Claude Code / Swe-Agent) with iterative execution feedback would clarify whether the performance drop reflects fundamental reasoning limits or merely recoverable brittleness.

3. **Limited root-cause analysis and mitigation.** While the benchmark excels at diagnosing failures, it offers minimal insight into resolving them. Including baseline mitigation experiments (e.g. few-shot prompting with interaction examples or supervised finetuning) would significantly elevate the paper's contribution beyond pure evaluation.

4. **Test coverage and circularity risks.** It is unclear if the fixed 100-test-case suite adequately covers tasks with massive parameter spaces, and the online agreement audit (only 5 submissions/task) is too sparse to guarantee robustness. Furthermore, using GPT-5.2 and Gemini-3-Pro to both construct the interactors and serve as evaluation targets introduces potential leniency bias toward their own coding patterns.

---

> ### Author Rebuttal · Authors · 2026-03-31
>
> We thank the reviewer for the constructive questions and for recognizing the ''well-motivated gap'', ''robust evaluation'' harness, and ''actionable failure modes''. We especially appreciate the controlled batch-vs.-interactive suggestion. We hope the new experiments below adequately resolve the remaining concerns.
>
> > **W1: Confounding between interaction and difficulty**
>
> **A:** This is a key concern, and we are grateful it was raised. We have run a controlled experiment on 150 problems, where the only difference is that the batch condition reveals the hidden test case through standard input.
>
> Revealing hidden information improves pass@1 by 0.037–0.256 across models (e.g., GPT-5.2: 0.767 vs. 0.511), indicating that information restriction is itself a substantial source of difficulty beyond the algorithmic core. The failure composition shifts accordingly: PE and QLE largely disappear in batch, while WA rises to dominate (0.69–0.80 of failures, up from 0.54–0.69 in interactive). An interesting rank reversal also appears — Qwen3-14B-NonThinking outperforms Qwen3-14B in batch (0.180 vs. 0.093) but underperforms in interactive (0.011 vs. 0.056). One possible explanation is that the thinking model allocates effort to information-acquisition strategies that become redundant when full input is available; in interactive settings, this same capability becomes an advantage.
>
> The full comparison is in Table 3 ([link](https://bit.ly/4s5lw6E)).
>
> > **W2, W3: Agentic evaluation and mitigation experiments**
>
> **A:** We agree that agentic evaluation is an important missing piece. We have conducted both iterative refinement (refine@5, following ICPC-Eval [1]) and few-shot prompting experiments across all 322 problems and five models.
>
> Surface-level errors are indeed recoverable with feedback: CE and PE drop substantially across rounds (e.g., GPT-5.2 CE: 0.040 → 0.010). However, WA persists as the dominant failure mode, and the overall gain is strongly capability-dependent: GPT-5.2 improves by +0.146, while Qwen3-14B-NonThinking gains only +0.010.
>
> Few-shot prompting with a solved interactive example tells a complementary story. It primarily helps weaker models with protocol handling (Qwen3-14B-NonThinking: 0.037 → 0.056) but has a negligible effect on stronger models (GPT-5.2: 0.705 → 0.714). Together, these results suggest that while some interaction errors are recoverable, the dominant failure mode, WA, reflects a reasoning limitation under partial observability that simple interventions alone cannot resolve. Notably, among the strategies we tested, independent resampling (pass@5) gives the strongest overall gains, likely because it explores more diverse solution trajectories.
>
> Per-round and per-strategy breakdowns are in Tables 1 and 2 ([link](https://bit.ly/4s5lw6E)).
>
> > **W4a, Q1: Test case sufficiency**
>
> **A:** Thank you for the detailed questions. We hope the following helps address both aspects.
>
> 1. **Test-suite sufficiency.** A bootstrap subsampling analysis (subset sizes 5–80 from the 100-case suite, 1,000 draws, five models) shows that pass@5 estimates stabilize well before the full suite size, and model ranking remains unchanged. Meaningful inflation appears only for very small suites (≤10). The stability analysis is in Figure 1 ([link](https://bit.ly/4s5lw6E)).
>
> 2. **Online agreement audit.** The online audit is a supplementary sanity check, not the primary validation mechanism. Interactor correctness is enforced by the verification pool: each candidate must match the official judge on all 45 execution-based checks (15 accepted, 15 rejected, and 15 mutation-derived submissions); only 100%-match candidates are accepted. The online audit cross-checks a sample of verdicts against the official judge for environment consistency; its scale is limited because platforms such as Codeforces prohibit automated submissions. We have clarified this in the revision and appreciate the reviewer's diligence in raising it.
>
> > **W4b, Q2: Leniency bias (circularity)**
>
> **A:** Thank you for raising this concern — circularity is indeed worth careful examination. As noted above, interactor acceptance is purely execution-based, so the process does not rely on subjective model judgment. To further verify this, we have implemented a leave-one-out evaluation on 154 tasks where both proposers independently passed the verification pool. GPT-5.2 was evaluated only on Gemini-authored interactors, and vice versa:
>
> | Model | pass@5 (original) | pass@5 (leave-one-out) |
> |-|-|-|
> | GPT-5.2 | 0.714 | 0.718 |
> | Gemini-3-Pro-Preview | 0.761 | 0.754 |
>
> The differences stay within 0.007, suggesting that proposer identity does not meaningfully affect evaluation outcomes. We hope this alleviates the circularity concern.
>
> [1] Shiyi Xu, Yiwen Hu, Yingqian Min, Zhipeng Chen, Wayne Xin Zhao, Ji-Rong Wen. *ICPC-Eval: Probing the Frontiers of LLM Reasoning with Competitive Programming Contests.*

---

> > ### Author Rebuttal · Reviewer_b2hV · 2026-04-03
> >
> > I thank the authors for their rebuttal and will maintain my positive score.

---

> > > ### Author Response · Authors · 2026-04-08
> > >
> > > We sincerely thank the reviewer for the positive assessment and encouraging feedback. We have incorporated all additions discussed in the rebuttal into the revised manuscript and would be grateful for any further support or suggestions to further strengthen the paper.

---

### Decision · Program_Chairs · 2026-04-30

**Decision:**

Accept (regular)

**Comment:**

This paper fills a real gap by turning interactive programming into a large-scale, well-validated benchmark rather than a small add-on setting. Reviewers found the benchmark design strong, the evaluation thorough, and the rebuttal convincing, especially on the controlled batch-vs-interactive comparison and the added agentic analyses.